# EFFECTS OF DATA GEOMETRY IN EARLY DEEP LEARNING

## ABSTRACT

Deep neural networks can approximate functions on different types of data, from images to graphs, with varied underlying structure. This underlying structure can be viewed as the geometry of the data manifold. By extending recent advances in the theoretical understanding of neural networks, we study how a randomly initialized neural network with piecewise linear activation splits the data manifold into *regions* where the neural network behaves as a linear function. We derive bounds on the number of linear regions and the distance to boundaries of these linear regions on the data manifold. This leads to insights into the expressivity of randomly initialized deep neural networks on non-Euclidean data sets. We empirically corroborate our theoretical results using a toy supervised learning problem. Our experiments demonstrate that number of linear regions varies across manifolds and how our results hold upon changing neural network architectures. We further demonstrate how the complexity of linear regions changes on the low dimensional manifold of images as training progresses, using the MetFaces dataset.

## 1 INTRODUCTION

The capacity of Deep Neural Networks (DNNs) to approximate arbitrary functions given sufficient training data in the supervised learning setting is well known (Cybenko, 1989; Hornik et al., 1989; Anthony & Bartlett, 1999). Several different theoretical approaches have emerged that study the effectiveness and pitfalls of deep learning. These studies vary in their treatment of neural networks and the aspects they study range from convergence (Allen-Zhu et al., 2019; Goodfellow & Vinyals, 2015), generalization (Kawaguchi et al., 2017; Zhang et al., 2017; Jacot et al., 2018; Sagun et al., 2018), function complexity (Montúfar et al., 2014; Mhaskar & Poggio, 2016), adversarial attacks (Szegedy et al., 2014; Goodfellow et al., 2015) to representation capacity (Arpit et al., 2017). Some recent theories have also been shown to closely match empirical observations (Poole et al., 2016; Hanin & Rolnick, 2019b; Kunin et al., 2020).

One approach for studying DNNs is to examine how the underlying structure, or geometry, of the data interacts with learning dynamics. The manifold hypothesis states that high-dimensional real world data typically lies on a low dimensional manifold (Tenenbaum, 1997; Carlsson et al., 2007; Fefferman et al., 2013). Studies have shown that DNNs are highly effective in deciphering this underlying structure by learning intermediate latent representations (Poole et al., 2016). The ability of DNNs to "flatten" complex data manifolds, using composition of seemingly simple piece-wise linear functions, appears to be unique (Brahma et al., 2016; Hauser & Ray, 2017).

DNNs with piecewise linear activations, such as ReLU (Nair & Hinton, 2010), divide the input space into linear regions, wherein the DNN behaves as a linear function (Montúfar et al., 2014). The density of these linear regions serves as a proxy for the DNN's ability to interpolate a complex data landscape and has been the subject of detailed studies (Montúfar et al., 2014; Telgarsky, 2015; Serra et al., 2018; Raghu et al., 2017). The work by Hanin & Rolnick (2019a) on this topic stands out because they derive bounds on the average number of linear regions, as opposed to worst case bounds, and verify the tightness of these bounds empirically for deep ReLU networks. Hanin & Rolnick (2019a) conjecture that the number of linear regions correlates to the expressive power of randomly initialized DNNs with piecewise linear activations. However, they assume that the data is uniformly sampled from the Euclidean space $\mathbb{R}^d$, for some $d$. By combining the manifold hypothesis with insights from Hanin & Rolnick (2019a), we are able to go further in estimating the the number of linear regions

and the average distance from *linear boundaries*. We derive bounds on how the geometry of the data manifold affects the aforementioned quantities.

To corroborate our theoretical bounds with empirical results, we design a toy problem where the input data is sampled from two distinct manifolds that can be represented in a closed form. We count the exact number of linear regions on these two manifolds that a randomly initialized neural network splits them into. We also observe the average distance to the boundaries of linear regions. We demonstrate how the number of linear regions varies for two distinct manifolds in our setting. These results show that the number of linear regions on the manifold do not grow exponentially with the dimension of input data. Our experiments do not provide estimates for theoretical constants, as is the case in deep learning theory, but demonstrate that the number of linear regions change as a consequence of these constants. We also study high dimensional data that lies on low dimensional manifolds with unknown structure and how the number of linear regions vary on and off this manifold, which is a more realistic setting. To achieve this we present experiments performed on the manifold of natural images. We sample data from the image manifold using a generative adversarial network (GAN) (Goodfellow et al., 2014) trained on the curated images of paintings. Specifically, we generate images using the pre-trained StyleGAN (Karras et al., 2019; 2020b) trained on the curated MetFaces dataset (Karras et al., 2020a). We also assign random labels to the images in the dataset and train a deep ReLU network in a supervised manner, a scenario in which it would overfit (Zhang et al., 2017). We generate *curves* on the image manifold of faces, using StyleGAN, and show how overfitting is reflected in the density of linear regions of the aforementioned deep ReLU network.

## 2 PRELIMINARIES AND BACKGROUND

Our goal is to understand how the underlying structure of real world data matters for deep learning. We first provide the mathematical background required to model this underlying structure as the geometry of data. We then provide a summary of previous work on understanding the approximation capacity of deep ReLU networks via the complexity of linear regions. For the details on how our work fits into the theory of DNNs we refer the reader to Appendix C.

### 2.1 DATA MANIFOLD AND DEFINITIONS

We use the example of the MetFaces dataset (Karras et al., 2020a) to illustrate how data lies on a low dimensional manifold. The images in the dataset are $1028 \times 1028 \times 3$ dimensional. By contrast, the number of *realistic* dimensions along which they vary are limited, e.g. painting style, artist, size and shape of the nose, jaw and eyes, background, clothing style; in fact, very few $1028 \times 1028 \times 3$ dimensional images correspond to realistic faces. We illustrate how this affects the possible variations in the data in Figure 1.

A manifold formalises the notion of limited variations in high dimensional data. One can imagine that there exists an unknown function $f : X \to Y$ from a low dimensional space of variations, to a high dimensional space of the actual data points. Such a function $f : X \to Y$, from one open subset $X \subset \mathbb{R}^m$, to another open subset $Y \subset R^k$, is a *diffeomorphism* if $f$ is bijective, and both $f$ and $f^{-1}$ are differentiable, also referred to as smooth. Therefore, a manifold is defined as follows.

**Definition 2.1.** *Let $k, m \in \mathbb{N}_0$. A subset $M \subset \mathbb{R}^k$ is called a smooth $m$-dimensional submanifold of $\mathbb{R}^k$ (or $m$-manifold in $\mathbb{R}^k$) iff every point $x \in M$ has an open neighborhood $U \subset \mathbb{R}^k$ such that $U \cap M$ is diffeomorphic to an open subset $\Omega \subset \mathbb{R}^m$. A diffeomorphism (i.e. differentiable mapping),*

$$f : U \cap M \to \Omega$$

*is called a coordinate chart of M and the inverse,*

$$h := f^{-1} : \Omega \to U \cap M$$

*is called a smooth parametrization of $U \cap M$.*

For the MetFaces dataset example, suppose there are 10 dimensions along which the images vary. Further assume that each variation can take a value continuously in some interval of $\mathbb{R}$. Then the smooth parametrization would map $f : \Omega \cap \mathbb{R}^{10} \to M \cap \mathbb{R}^{1028 \times 1028 \times 3}$. This parametrization and its inverse are unknown in general, and computationally very difficult to estimate in practice.

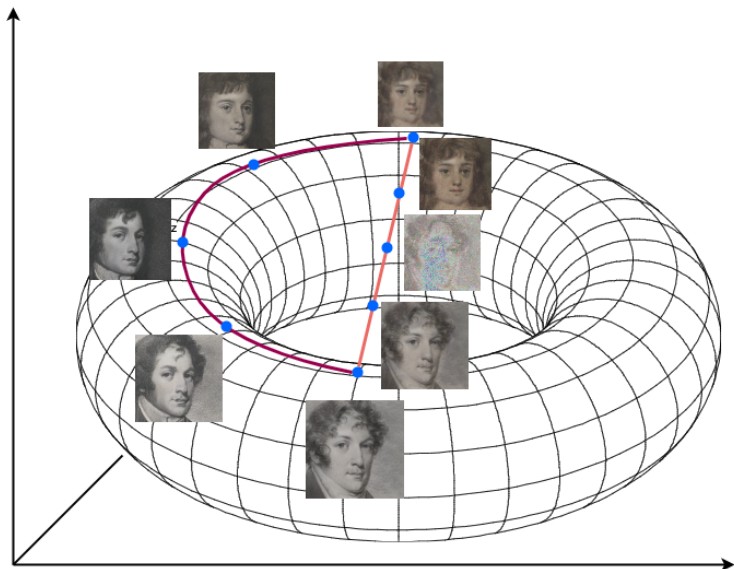

Figure 1: A visualization of how the 2D surface, here represented by a 2-torus, is embedded in a larger input space, $R^3$. Suppose each point corresponds to an image of the face on this 2-torus. We can chart two curves: one straight line cutting across the 3D space and another curve that stays on the torus. The images corresponding to the points on the torus will have a smoother variation in style and shape whereas there will be images corresponding to points on the straight line that do not belong to the class of pictures of faces.

There are similarities in how geometric elements are defined for manifolds and Euclidean spaces. A smooth curve, on a manifold $M$, $\gamma : I \to M$ is defined from an interval $I$ to the manifold $M$ as a function that is differentiable for all $t \in I$, just as is done for Euclidean spaces. The shortest such curve between two points on a manifold is no longer a straight line, but is instead a *geodesic*. One recurring geometric element, which is unique to manifolds and stems from the definition of smooth curves, is that of a *tangent space*, defined as follows.

**Definition 2.2.** *Let $M$ be an $m$-manifold in $\mathbb{R}^k$ and $x \in M$ be a fixed point. A vector $v \in \mathbb{R}^k$ is called a tangent vector of $M$ at $x$ if there exists a smooth curve $\gamma : I \to M$ such that $\gamma(0) = x, \dot{\gamma}(0) = v$ where $\dot{\gamma}(t)$ is the derivative of $\gamma$ at $t$. The set*

$$T_x M := \{\dot{\gamma}(0) | \gamma : \mathbb{R} \to M \text{ is smooth} \gamma(0) = x\},$$

*of tangent vectors of $M$ at $x$ is called the tangent space of $M$ at $x$.*

In simpler terms, the plane tangent to the manifold $M$ at point $x$ is called the tangent space and denoted by by $T_x M$. Consider the upper half of a 2-sphere, $S^2 \subset \mathbb{R}^3$, which is a 2-manifold in $\mathbb{R}^3$. The tangent space at a fixed point $x \in S^2$ is the 2D plane perpendicular to the vector $x$ and tangential to the surface of the sphere that contains the point $x$. We refer to these concepts in the text that follows. For additional background on manifolds we refer the reader to Appendix B.

## 2.2 LINEAR REGIONS OF DEEP RELU NETWORKS

We consider a neural network, $F$, which is a composition of activation functions. Inputs at each layer are multiplied by a matrix, referred to as the weight matrix, with an additional bias vector that is added to this product. We limit our study to ReLU activation function (Nair & Hinton, 2010), which is piece-wise linear and one of the most popular activation functions being applied to various learning tasks on different types of data like text, images, signals etc. We further consider DNNs that map inputs, of dimension $n_{\text{in}}$, to scalar values, i.e. values in $\mathbb{R}$. Therefore, $F : \mathbb{R}^{n_{\text{in}}} \to \mathbb{R}$ is defined as,

$$F(x) = W_L \sigma(B_{L-1} + W_{L-1} \sigma(...\sigma(B_1 + W_1 x))), \tag{1}$$

where $W_l \in \mathbb{M}^{n_l \times n_{l-1}}$ is the weight matrix for the $l^{\text{th}}$ hidden layer, $n_l$ is the number of neurons in the $l^{\text{th}}$ hidden layer, $B_l \in \mathbb{R}^{n_l}$ is the vector of biases for the $l^{\text{th}}$ hidden layer, $n_0 = n_{\text{in}}$ and $\sigma : \mathbb{R} \to \mathbb{R}$

is the activation function. For a neuron $z$ in the $l^{\text{th}}$ layer we denote the *pre-activation* of this neuron, for given input $x \in \mathbb{R}^{n_{\text{in}}}$, as $z_l(x)$. For a neuron $z$ in the layer $l$ we have

$$z(x) = W_{l-1,z}\sigma(...\sigma(B_1 + W_{1,z}x)),$$

for $l > 1$ (for the base case $l = 1$ we have $z(x) = W_{1,z}x$) where $W_{l-1,z}$ is the row of weights, in the weight matrix of the $l^{\text{th}}$ layer, $W_l$, corresponding to the neuron $z$. We use $W_z$ to denote the weight vector for brevity, omitting the layer index $l$ in the subscript. We also use $b_z$ to denote the bias term for the neuron $z$.

Neural networks with piecewise linear activations are piecewise linear on the input space (Montúfar et al., 2014). Suppose for some fixed $y \in \mathbb{R}^{n_{\text{in}}}$ as $x \to y$ if we have $z(x) \to -b_z$ then we observe a discontinuity in the gradient $\nabla_x \sigma(b_z + W_z z(x))$ at $y$. Intuitively, this is because $x$ is approaching the boundary of the linear region of the function defined by the output of $z$. Therefore, the boundary of linear regions, for a feed forward neural network $F$, is defined as:

$$\mathcal{B}_F = \{x | \nabla F(x) \text{ is not continuous at } x\}.$$

Hanin & Rolnick (2019a) argue that an important generalization for the approximation capacity of a neural network $F$ is the $(n_{\text{in}} - 1)-$dimensional volume density of linear regions defined as $\text{vol}_{n_{\text{in}}-1}(\mathcal{B}_F \cap K)/\text{vol}_{n_{\text{in}}}(K)$, for a bounded set $K \subset \mathbb{R}^{n_{\text{in}}}$. This quantity serves as a proxy for density of linear regions and therefore the expressive capacity of DNNs. Intuitively, higher density of linear boundaries means higher capacity of the DNN to approximate complex non-linear functions. The quantity is applied to lower bound the distance between a point $x \in K$ and the set $\mathcal{B}_F$, which is

$$\text{distance}(x, \mathcal{B}_F) = \min_{\text{neurons } z} |z(x) - b_z|/||\nabla z(x)||,$$

which measures the sensitivity over neurons at a given input. The above quantity measures how "far" the input is from flipping any neuron from inactive to active or vice-versa.

Informally, Hanin & Rolnick (2019a) provide two main results for a randomly initialized DNN $F$, with a reasonable initialisation. Firstly, they show that

$$\mathbb{E}\Big[\frac{\text{vol}_{n_{\text{in}}-1}(\mathcal{B}_F \cap K)}{\text{vol}_{n_{\text{in}}}(K)}\Big] \approx \#\{ \text{ neurons}\},$$

meaning the density of linear regions is bound above and below by some constant times the number of neurons. Secondly, for $x \in [0,1]^{n_{\text{in}}}$,

$$\mathbb{E}\Big[\text{distance}(x, \mathcal{B}_F)\Big] \geq C\#\{ \text{ neurons}\}^{-1},$$

where $C > 0$ depends on the distribution of biases and weights, in addition to other factors. Meaning that the distance to the nearest boundary is bounded above and below by a constant times the inverse of the number of neurons. These results stand in contrast to earlier worst case bounds that are exponential in the number of neurons. Hanin & Rolnick (2019a) also verify these results empirically to note that the constants lie in the vicinity of 1 throughout training.

## 3 LINEAR REGIONS ON THE DATA MANIFOLD

One important assumption in the results presented by Hanin & Rolnick (2019a) is that the input, $x$, lies in a compact set $K \subset \mathbb{R}^{n_{\text{in}}}$ and that $\text{vol}_{n_{\text{in}}}(K)$ is greater than 0. Also, the theorem pertaining to the lower bound on average distance of $x$ to linear boundaries the input assumes the input uniformly distributed in $[0,1]^{n_{\text{in}}}$. As noted earlier, high-dimensional real world datasets, like images, lie on low dimensional manifolds, therefore both these assumptions are false in practice. Therefore, we study the case where the data lies on some $m-$dimensional submanifold of $\mathbb{R}^{n_{\text{in}}}$, i.e. $M \subset \mathbb{R}^{n_{\text{in}}}$ where $m \ll n_{\text{in}}$. We illustrate how this additional constraint effects the study of linear regions in Figure 2.

As introduced by Hanin & Rolnick (2019a), we denote the "$(n_{\text{in}} - k)-$dimensional piece" of $\mathcal{B}_F$ as $\mathcal{B}_{F,k}$. More precisely, $\mathcal{B}_{F,0} = \emptyset$ and $\mathcal{B}_{F,k}$ is recursively defined to be the set of points $x \in \mathcal{B}_F \setminus \{\mathcal{B}_{F,0} \cup ... \cup \mathcal{B}_{F,k-1}\}$ with the added condition that in a neighbourhood of $x$ the set $\mathcal{B}_{F,k}$ coincides with hyperplane of dimension $n_{\text{in}} - k$. In our setting, where the data lies on a manifold $M$, we define $\mathcal{B}'_{F,k}$ as $\mathcal{B}_{F,k} \cap M$, and note that $\dim(\mathcal{B}'_{F,k}) = m - k$ (Appendix E Proposition 6). For

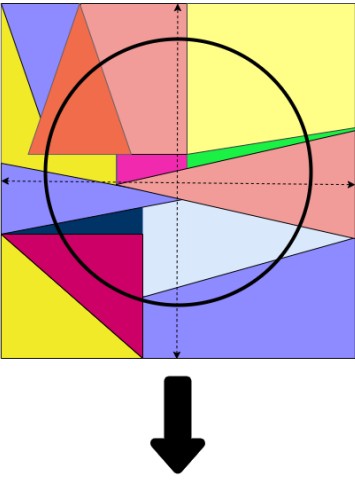

Figure 2: A circle is an example of a 1D manifold in a 2D Euclidean space. The effective number of linear regions on the manifold, the upper half of the circle, are the number of linear regions on the arc from $-\pi$ to $\pi$. In the diagram above, each color in the 2D space corresponds to a linear region. When the upper half of the circle is flattened into a 1D space we obtain a line. Each color on the line corresponds to a linear region of the 2D space.

example, the *transverse* intersection (see Definition E.1) of a plane in 3D with the 2D manifold $S^2$ is a 1D curve in $S^2$ and therefore has dimension 1. Therefore, $\mathcal{B}'_{F,k}$ is a submanifold of dimension $3 - 2 = 1$. This imposes the restriction $k \leq m$, for the intersection $\mathcal{B}_{F,k} \cap M$ to have a well defined volume.

We first note that the definition of the determinant of the Jacobian, for a collection of neurons $z_1, ..., z_k$, is different in the case when the data lies on a manifold $M$ as opposed to in a compact set of dimension $n_{\text{in}}$ in $\mathbb{R}^{n_{\text{in}}}$. Since the determinant of the Jacobian is the quantity we utilise in our proofs and theorems repeatedly we will use the term Jacobian to refer to it for succinctness. Intuitively, this follows from the Jacobian of a function being defined differently in the ambient space $\mathbb{R}^{n_{\text{in}}}$ as opposed to the manifold $M$. In case of the former it is the volume of the paralellepiped determined by the vectors corresponding to the directions with steepest ascent along each one of the $n_{\text{in}}$ axes. In case of the latter it is more complex and defined below. Let $\mathcal{H}^m$ be the $m-$dimensional Hausdorff measure (we refer the reader to the Appendix B for background on Hausdorff measure). The Jacobian of a function on manifold $M$, as defined by Krantz & Parks (2008) (Chapter 5), is as follows

**Definition 3.1.** *The (determinant of) Jacobian of a function $H : M \to \mathbb{R}^k$, where $k \leq \dim(M) = m$, is defined as*

$$J^M_{k,H}(x) = \sup \left\{ \frac{\mathcal{H}^k(D_M H(P))}{\mathcal{H}^k(P)} \middle| P \text{ is a } k\text{-dimensional parallelepiped contained in } T_x M. \right\},$$

*where $D_M : T_x M \to \mathbb{R}^k$ is the differential map (see Appendix B) and we use $D_M H(P)$ to denote the mapping of the set $P$ in $T_x M$, which is a parallelepiped, to $\mathbb{R}^k$. The supremum is taken over all parallelepipeds $P$.*

We also say that neurons $z_1, ..., z_k$ are good at $x$ if there exists a path of neurons from $z$ to the output in the computational graph of $F$ so that each neuron is activated along the path. Our three main results that hold under the assumptions listed in Appendix A, each of which extend and improve upon the theoretical results by Hanin & Rolnick (2019a), are:

**Theorem 1.** *Given $F$ a feed-forward ReLU network with input dimension $n_{in}$, output dimension $1$, and random weights and biases. Then for any bounded measurable submanifold $M \subset \mathbb{R}^{n_{in}}$ and any $k = 1, ...., m$ the average $(m - k)-$dimensional volume of $\mathcal{B}_{F,k}$ inside $M$,*

$$\mathbb{E}[vol_{m-k}(\mathcal{B}_{F,k} \cap M)] = \sum_{\text{distinct neurons } z_1, ..., z_k \text{ in } F} \int_M \mathbb{E}[Y_{z_1, ..., z_k}] dvol_m(x), \quad (2)$$

*where $Y_{z_1, ..., z_k}$ is $J^M_{m,H_k}(x)\rho_{b_1, ..., b_k}(z_1(x), ..., z_k(x))$, times the indicator function of the event that $z_j$ is good at $x$ for each $j = 1, ..., k$. Here the function $\rho_{b_{z_1}, ..., b_{z_k}}$ is the density of the joint distribution of the biases $b_{z_1}, ..., b_{z_k}$.*

This change in the formula, from Theorem 3 by Hanin & Rolnick (2019a), is a result of the fact that $z(x)$ has a different direction of steepest ascent when it is restricted to the data manifold $M$, for any $j$. The proof is presented in Appendix E. Formula 2 also makes explicit the fact that the data manifold has dimension $m \leq n_{\text{in}}$ and therefore the $m-k$-dimensional volume is a more representative measure of the linear boundaries. Equipped with Theorem 1, we provide a result for the density of linear regions on manifold $M$.

**Theorem 2.** *For data sampled uniformly from a compact and measurable $m$ dimensional manifold $M$ we have the following result for all $k \leq m$:*

$$\frac{vol_{m-k}(\mathcal{B}_{F,k} \cap M)}{vol_m(M)} \leq \binom{\# \; neurons}{k} (2C_{grad}C_{bias}C_M)^k,$$

*where $C_{grad}$ depends on $||\nabla z(x)||$ and the DNN's architecture, $C_M$ depends on the geometry of $M$, and $C_{bias}$ on the distribution of biases $\rho_b$.*

The constant $C_M$ is the supremum over the matrix norm of projection matrices onto the tangent space, $T_x M$, at any point $x \in M$. For the Euclidean space $C_M$ is always equal to 1 and therefore the term does not appear in the work by Hanin & Rolnick (2019a), but we cannot say the same for our setting. We refer the reader to Appendix F for the proof, further details, and interpretation. Finally, under that added assumptions that the diameter of the manifold $M$ is finite and $M$ has polynomial volume growth we provide a lower bound on the average distance to the linear boundary for points on the manifold and how it depends on the geometry and dimensionality of the manifold.

**Theorem 3.** *For any point, $x$, chosen randomly from $M$, we have:*

$$\mathbb{E}[distance_M(x, \mathcal{B}_F \cap M)] \geq \frac{C_{M,\kappa}}{C_{grad}C_{bias}C_M \# neurons}$$

*where $C_{M,\kappa}$ depends on the scalar curvature and dimensionality of the manifold $M$. The function $distance_M$ is the distance on the manifold $M$.*

This result gives us intuition on how the density of linear regions around a point depends on the geometry of the manifold. Note that the constant $C_M$ is the same as in Theorem 2. Another difference to note is that we derive a lower bound on the geodesic distance on the manifold $M$ and not the Euclidean distance in $\mathbb{R}^k$ as done by Hanin & Rolnick (2019a). This distance better captures the distance between data points on a manifold while incorporating the underlying structure. In other words, this distance can be understood as how much a data point should change to reach a linear boundary while ensuring that all the individual points on the curve, tracing this change, are "valid" data points. We provide proof for the above theorem in Appendix G. For background on curvature of manifolds and a proof sketch we refer the reader to the Appendices B and D, respectively.

### 3.1 Intuition For Theoretical Results

One of the key ingredients of the proofs by Hanin & Rolnick (2019a) is the co-area formula.[1] The co-area formula is applied to get a closed form representation of the $k-$dimensional volume of the region where any set of $k$ neurons, $z_1, z_2, ..., z_k$ is "good" in terms of the expectation over the Jacobian, in the Euclidean space. Instead of the co-area formula we use the smooth co-area formula, [2] to get a closed form representation of the $m - k-$dimensional volume of the region intersected with manifold, $M$, in terms of the Jacobian defined on a manifold (Definition 3.1). The key difference between the two formulas is that in the smooth co-area formula the Jacobian (of a function from the manifold $M$) is "restricted" to the tangent plane. While the determinant of the vanilla Jacobian measures the distortion of volume around a point in Euclidean space the determinant of the Jacobian defined as above (Definition 3.1) measures the distortion of volume on the manifold instead for the function with the same domain, the function that is 1 if the set of neurons are good and 0 otherwise.

The value of the determinant as defined in 3.1 has the same volume as the projection of the parallelepiped defined by the gradients $\nabla z(x)$ onto the tangent space (see Proposition 8). This introduces the constant $C_M$, defined above. Essentially, the constant captures how the magnitude of the gradients, $\nabla z(x)$, are modified upon being projected to the tangent plane. Certain manifolds "shrink"

---

[1] https://en.wikipedia.org/wiki/Coarea_formula
[2] https://en.wikipedia.org/wiki/Smooth_coarea_formula

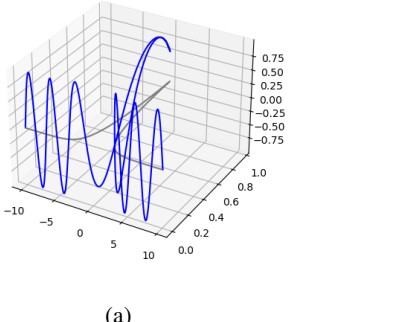 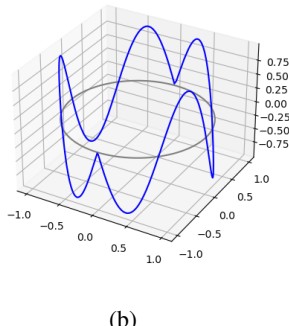

(a)                                              (b)

Figure 3: The tractrix (a) and circle (b) are plotted in grey on the x-y plane, which are the 1D input data manifolds. The target function is in blue, on the z-axis, and periodic in nature.

vectors upon projection to the tangent plane more than others, on an average, which is a function of their geometry. We illustrate of how two distinct manifolds "shrink" the gradients differently upon projection to the tangent plane and it is reflected in the number of linear regions on the manifolds (see Figure 11 in the appendix). We provide intuition for the curvature of a manifold in Appendix B, due to space constraints, which is used in the lower bound for the average distance in Theorem 3.

## 4 EXPERIMENTS

### 4.1 SUPERVISED LEARNING ON TOY DATASET

To empirically corroborate our theoretical results, we calculate the number of linear regions and average distance to the linear boundary, bounds for which are presented in the theorems above, on a regression task for two different manifolds. To achieve this we define two similar regression tasks where the data is sampled from two different manifolds with different geometries. We parameterize the first task, a unit circle without its north and south poles, by $\psi_{\text{circle}} : (-\pi, \pi) \to \mathbb{R}^2$ where $\psi_{\text{circle}}(\theta) = (\cos \theta, \sin \theta)$ and $\theta$ is the angle made by the vector from the origin to the point with respect to the x-axis. We set the target function for regression task to be a periodic function in $\theta$. The target is defined as $z(\theta) = a \sin(\nu \theta)$ where $a$ is the amplitude and $\nu$ is the frequency (Figure 3). DNNs have difficulty learning periodic functions (Ziyin et al., 2020). The motivation behind this is to present the DNN with a challenging task where it has to learn the underlying structure of the data. Moreover the DNN will have to split the circle into linear regions. For the second regression task, a tractrix is parametrized by $\psi_{\text{tractrix}} : \mathbb{R}^1 \to \mathbb{R}^2$ where $\psi_{\text{tractrix}}(y) = (t - \tanh t, \operatorname{sech} t)$ (see Figure 3). We assign a target function $z(t) = a \sin(\nu t)$. For the purposes of our study we restrict the domain of $\psi_{\text{tractrix}}$ to $(-3, 3)$. This allows us to observe effects of varying data geometry across the manifolds.

The results, averaged over 20 runs, are presented in Figures 4 and 5. We note that $C_M$ is smaller for Sphere (based on Figure 4) and the curvature is positive whilst $C_M$ is larger for tractrix and the curvature is negative. Both of these constants (curvature and $C_M$) contribute to the lower bound in Theorem 3. Similarly, we show results of number of linear regions divided by the number of neurons upon changing architectures, consequently the number of neurons, for the two manifolds in Figure 8, averaged over 30 runs. Note that this experiment observes the effect of $C_M \times C_{\text{grad}}$, since changing the architecture also changes $C_{\text{grad}}$. Although this variation in $C_{\text{grad}}$ is quite low in magnitude as observed empirically by Hanin & Rolnick (2019a). The empirical observations are consistent with our theoretical results. We observe that the number of linear regions starts off close to #neurons and remains close throughout the training process for both the manifolds. This supports our theoretical results (Theorem 2) that the constant $C_M$, which is distinct across the two manifolds, affects the number of linear regions throughout training. The tractrix has a higher value of $C_M$ and that is reflected in the results. This is due to different "shrinking" of vectors upon being projected to the tangent space as discussed in Section 3.1.

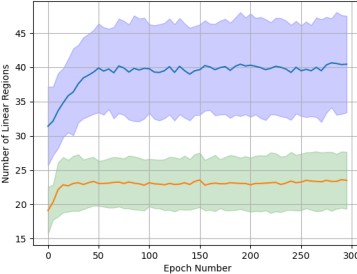

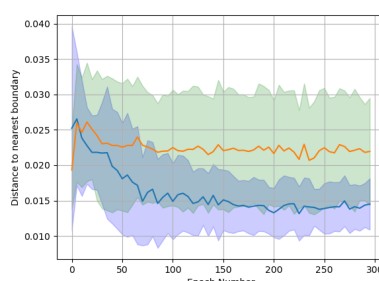

Figure 4: Graph of number of linear regions for tractrix (blue) and sphere (orange). The shaded regions represent one standard deviation. Note that the number of neurons is 26 and the number of linear regions is comparable but different for both the manifolds.

Figure 5: Graph of distance to linear regions for tractrix (blue) and sphere (orange). The distances are normalized by the maximum distance on the range, for both tractrix and sphere. The shaded regions represent one standard deviation.

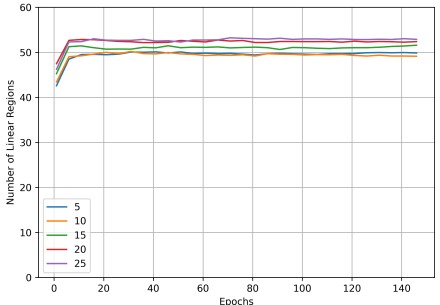

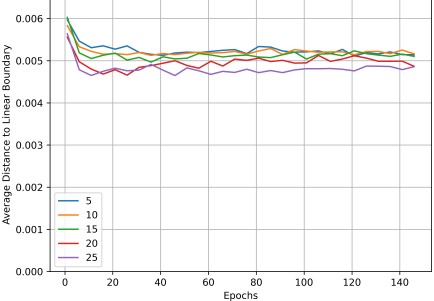

Figure 6: We observe that as the dimension $n_{\text{in}}$ is increased, while keeping the manifold dimension constant, the number of linear regions remains proportional to number of neurons (26).

Figure 7: We observe that as the dimension $n_{\text{in}}$ is increased, while keeping the manifold dimension constant, the average distance varies very little.

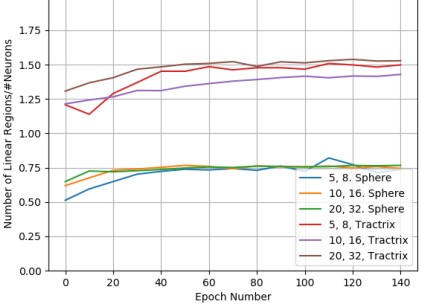

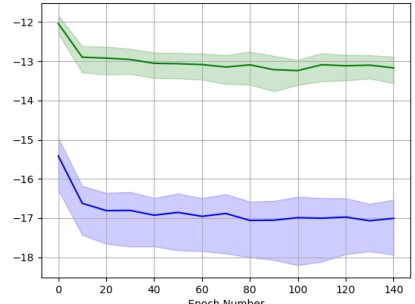

Figure 8: The effects of changing the architecture on the number of linear regions. We observe that the value of $C_M$ effects the number of linear regions proportionally. The number of hidden units for three layer networks are in the legend along with the data manifold.

Figure 9: We observe that the log density of number of linear regions is lower on the manifold (blue) as compared to off the manifold (green). As training progresses, in contrast to previous examples with generalization, the log density decreases.

### 4.2 VARYING $n_{\text{IN}}$

To empirically corroborate the results of Theorems 2 and 3 we vary the dimension $n_{\text{in}}$ while keeping $m$ constant. We achieve this by counting the number of linear regions and the average distance to boundary region on the 1D circle as we vary the input dimension in steps of 5. We draw samples of 1D circles in $\mathbb{R}^{n_{\text{in}}}$ by randomly choosing two perpendicular basis vectors. We then train a neural network with the same architecture as in the previous section on the periodic target function $(a \sin(\nu\theta))$ as defined above. The results in Figure 6 shows that the quantities stay proportional to $\#neurons$, and do not vary as $n_{\text{in}}$ is increased, as predicted by our theoretical results. This stands in contrast to the results by Hanin & Rolnick (2019a) where the upper and lower bounds both grow exponentially with $n_{\text{in}}$ for the number of linear regions in a compact set of $\mathbb{R}^{n_{\text{in}}}$. We provide implementation details in Appendix H.

### 4.3 METFACES: HIGH DIMENSIONAL DATASET

Our goal with this experiment is to study how overfitting relates to the number of linear regions of deep ReLU networks, in addition to observing the density of linear regions for very high dimensional image data that lies on a low dimensional manifold. To discover latent low dimensional underlying structure of data we employ a GAN. Adversarial training of GANs can be effectively applied to learn a mapping from a low dimensional latent space to high dimensional data (Goodfellow et al., 2014). The generator is a neural network that maps $g : \mathbb{R}^k \to \mathbb{R}^{n_{\text{in}}}$. Recently, Karras et al. (2019) introduced a new generator, StyleGAN, that interpolates better, meaning that it can disentangle the factors of variation in the dataset. As a follow up, Karras et al. (2020a) train the StyleGAN in a data efficient manner on the MetFaces dataset. We train a deep ReLU network on the MetFaces dataset with random labels (chosen from $0, 1$) with cross entropy loss. As noted by Zhang et al. (2017), training with random labels can lead to the DNN memorizing the entire dataset with poor generalization. Further implementation details are in Appendix I.

We compare the log density of number of linear regions on a curve on the manifold with a straight line off the manifold (see Figure 9). This leads to two observations: **1)** the density of the linear regions decreases as training progresses, in case of overfitting, which is in contrast to the scenario without overfitting (Hanin & Rolnick, 2019a) and it ties the pathological behavior of deep ReLU networks to density of linear regions, **2)** the density of linear regions is significantly lower on the data manifold and devising methods to "concentrate" these linear regions on the manifold is a promising research direction. That could lead to increased expressivity for the same number of parameters.

## 5 DISCUSSION AND FUTURE WORK

There is significant amounts of work in both supervised and unsupervised learning settings for non-Euclidean data (Bronstein et al., 2017). Despite these empirical results most theoretical analysis remains agnostic to data geometry, with a few prominent exceptions (Cloninger & Klock, 2020; Shaham et al., 2015; Chen et al., 2019; Schmidt-Hieber, 2019). We incorporate the idea of data geometry into measuring the effective approximation capacity of DNNs. We derive average bounds on the number of linear regions and distance from the linear boundary under the added assumption that the data is sampled from a low dimensional manifold. Our experimental results corroborate our theoretical results. We also present insights into overfitting in high dimensional datasets where the data lies on a low dimensional manifold. Estimating the geometry, dimensionality and curvature, of these image manifolds accurately is a problem that remains largely unsolved (Brehmer & Cranmer, 2020; Perraul-Joncas & Meila, 2013), which limits our inferences on high dimensional dataset to observations that guide future research. We note that proving a lower bound on the number of linear regions, as done by Hanin & Rolnick (2019a), for the manifold setting remains open. Our work opens up avenues for further research that combines model geometry and data geometry and can lead to empirical research geared towards developing DNN architectures for high dimensional datasets that lie on a low dimensional manifold.

## 6 REPRODUCIBILITY STATEMENT

We offer the following details for all the experiments we have performed: hyperparameters, sample sizes, GPU-hours, CPU-hours, code, neural network architectures, python libraries, input sizes, and external code bases. We also provide the code with instructions on running it in the `readme.txt` in the `exp/` folder of the supplementary material. Specifically, for Sections 4.1 and 4.2 the implementation details are in Appendix H, Section 4.3 in Appendix I. We also provide additional details for running the StyleGan2 code and its license in Appendix J.

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

## A  ASSUMPTIONS

We first make explicit the assumptions on the distribution of weights and biases.

**A1:** The conditional distribution of any set of biases $b_{z_1}, ..., b_{z_k}$ given all other weights and biases has a density $\rho_{z_1,...,z_k}(b_1, ..., b_k)$ with respect to Lebesgue measure on $\mathbb{R}^k$.

**A2:** The joint distribution of all weights has a density with respect to Lebesgue measure on $\mathbb{R}^{\#\text{weights}}$.

**A3:** The data manifold $M$ is smooth.

**A4:** (Only needed for Theorem 3) the diameter of $M$ defined by $d_M = \sup_{x,y \in M} \text{distance}_M(x, y)$ is finite.

**A5:** (Only needed for Theorem 3) a geodesic ball in manifold $M$ has polynomial volume growth of order $m$.

## B ADDITIONAL BACKGROUND ON MANIFOLDS

We provide further background on the theory of manifolds. In this section we first provide the background, definition and an interpretation for the **scalar curvature** of a manifold at a point. Every smooth manifold is also equipped with a *Riemannian metric tensor* (or metric tensor in short). Given any two vectors, $v$ and $w$, in the tangent space of a point $x$ on a manifold $M$, the metric tensor defines a parallel to the dot product in Euclidean spaces. The metric tensor, at a point $x$, is defined by the smooth functions $g_{ij} : M \to \mathbb{R}, i, j \in \{1, ..., k\}$. Where the matrix defined by

$$ G_x = [g_{ij}(x)] = \begin{bmatrix} g_{11}(x) & \dots & g_{1n}(x) \\ \vdots & \ddots & \vdots \\ g_{n1}(x) & \dots & g_{nn}(x) \end{bmatrix} $$

is symmetric and invertible. The inner product of $u, v \in T_x M$ is then defined by $\langle u, v \rangle_M = u^T G_x v$. the inner product is symmetric, non-degenerate, and bilinear, i.e.

$$ \langle ku, v \rangle_M = k \langle u, v \rangle_M = \langle u, kv \rangle_M, $$
$$ \langle u + w, v \rangle_M = \langle u, v \rangle_M + \langle w, v \rangle_M, $$
$$ \langle u, v \rangle_M = \langle v, u \rangle_M. $$

As can be seen, these properties also hold for the Euclidean inner product (with $G_x = I$ for all $x$). Let the inverse of $G = [g_i j(x)]$ be denoted by $[g^{ij}(x)]$. Building on this definition of the metric tensor the Ricci curvature tensor is defined as

$$ R_{ij} = -\frac{1}{2} \sum_{a,b=1}^{n} \left( \frac{\partial^2 g_{ij}}{\partial x_a \partial x_b} + \frac{\partial^2 g_{ab}}{\partial x_i \partial x_j} - \frac{\partial^2 g_{ib}}{\partial x_j \partial x_a} - \frac{\partial^2 g_{jb}}{\partial x_i \partial x_a} \right) g^{ab} $$
$$ + \sum_{a,b,c,d=1}^{n} \left( \frac{1}{2} \frac{\partial g_{ac}}{\partial x_i} \frac{\partial g_{bd}}{\partial x_j} + \frac{\partial g_{ic}}{\partial x_a} \frac{\partial g_{jd}}{\partial x_b} - \frac{\partial g_{ic}}{\partial x_a} \frac{\partial g_{jb}}{\partial x_d} \right) g^{ab} g^{cd} $$
$$ - \frac{1}{4} \sum_{a,b,c,d=1}^{n} \left( \frac{\partial g_{jc}}{\partial x_i} + \frac{\partial g_{ic}}{\partial x_j} - \frac{\partial g_{ij}}{\partial x_c} \right) g^{ab} g^{cd}. $$

For geometric interpretations of the above tensors we refer the reader to the work by Loveridge (2004).

Another quantity, from the theory of manifolds, which we utilise in our proofs and theorems, is scalar curvature (or Ricci curvature). The curvature is a measure how much the volume of a geodisic ball on the manifold M, e.g. $S^2$, deviates from a $d - 1$ sphere in the flat space, e.g. $\mathbb{R}^3$. The volume on the manifold deviates by an amount proportional to the curvature. We illustrate this idea in figure 10. We refer the reader to works by Gray (1974) and Wan (2016) for further technical details. Since our main theorems relate to the volume of linear regions the scalar curvature plays an important role. Formally, the scalar curvature of a manifold $M$ at a point $x$ with metric tensor $[g_{ij}]$ and Ricci tensor $[R_{ij}]$ is defined as

$$ C = \sum_{i,j=1}^{n} g^{ij} R_{ij}. $$

Another important concept is that of **Hausdorff measure**. Since the volumes are "distorted" on a manifold it requires careful consideration when defining a measure and integrating using it on a manifold. The $m-$dimensional Hausdorff measure, of a set $S$, is defined as

$$ H^m(S) := \sup_{\delta > 0} \inf \left\{ \sum_{i=1}^{\infty} (\text{diam } U_i)^d | S \subseteq \cup_{i=1}^{\infty} U_i, \text{diam } U_i < \delta \right\}. $$

Next we introduce the definition of the **differential map** that is used in Definition 3.1, for the determinant of the Jacobian. The differential map of a smooth function $H$ from a manifold $M$ to a manifold $S$ at a point $x \in M$ is the smooth map $dH : T_x M \to T_x S$ such that the tangent vector corresponding to any smooth curve $\gamma : I \to M$ at $x$, $\gamma'(0) \in T_x M$, maps to the tangent vector of

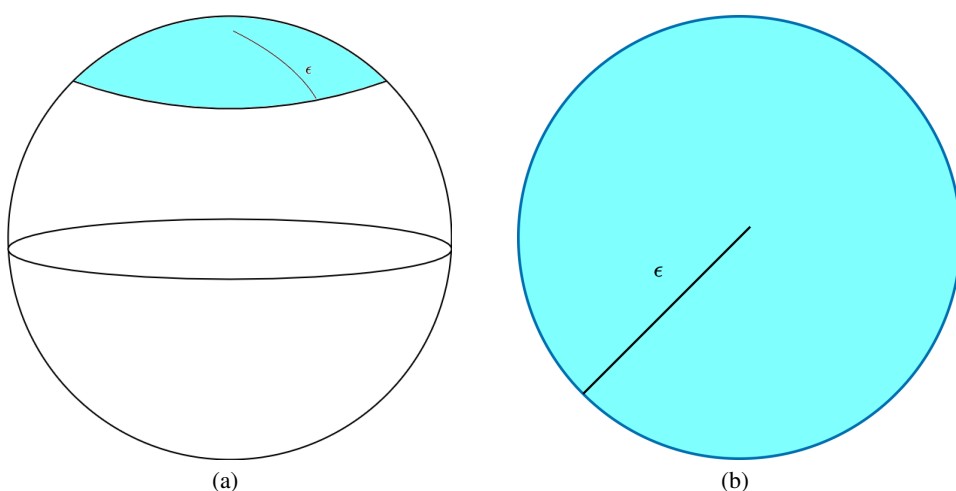

Figure 10: The geodesic circle on $S^2$ (blue region in (a)) does not have the same area as the flat circle (b), both of radius $\epsilon$. One can imagine cutting the blue top off the sphere's surface and trying to "flatten" it. Such an effort will lead to failure, if the material of the sphere does not "stretch", since the geodesic ball, on $S^2$, cannot be mapped to a circle in $\mathbb{R}^2$ in a distance preserving manner. Thus, the area of the two blue regions in (a) and (b) vary. This deviation in the area spanned by the two spheres, despite their radii being the same, is proportional to the scalar curvature.

$H \circ \gamma$ in $T_{H(x)}N$. This is the analog of the total derivative of "vanilla calculus". More intuitively, the differential map captures how the function changes along different directions on $N$ as its input changes along different directions on $M$, this also has an analog to how rows of the Jacobian matrix are viewed in calculus. In Definition 3.1 we use the specific case where the function $H$ maps from manifold $M$ to the Euclidean space $\mathbb{R}^k$ and the tangent space of a Euclidean space is the Euclidean space itself. Finally, a paralellepiped's, $P$ in $T_x M$, mapping via the differential map gives us the points in $\mathbb{R}^k$ that correspond to this set $P$.

## C   RELATED WORK

There have been various approaches to explain the efficacy of DNNs in approximating arbitrarily complex functions. We briefly touch upon two such promising approaches. Broadly, the theory of DNNs can be viewed from two lenses: expressive power (Hornik et al., 1989; Bartlett et al., 1998; Poole et al., 2016; Raghu et al., 2017; Kawaguchi et al., 2017; Neyshabur et al., 2018; Hanin, 2019) and learning dynamics (Saxe et al., 2014; Su et al., 2016; Smith & Le, 2018; Jacot et al., 2018; Lee et al., 2019; Arora et al., 2019a;b). These approaches are not independent of one another but complementary. For example, Kawaguchi et al. (2017) argue theoretically how the family of DNNs generalize well despite the large capacity of the function class. Neyshabur et al. (2018) provide PAC-Bayes generalization bounds which are improved upon by Arora et al. (2018). Hanin (2019) shows that Deep ReLU networks of finite width can approximate any continuous, convex or smooth functions on a unit cube. These works look at DNNs from the lens of expressive power. More recently, there has been a surge in explaining how various algorithms arrive at these almost accurate function approximations by applying different theoretical models of DNNs. Jacot et al. (2018) provide results for convergence and generalization of DNNs in the infinite width limit by introducing a the neural tangent kernel (NTK). Hanin & Nica (2020) provide finite depth and width corrections for the NTK. Another line of work within the learning dynamics literature looks at implicit regularization that emerge from the learning algorithm and over-parametrised DNNs (Arora et al., 2019a;b; Du et al., 2018; Liang et al., 2019).

Researchers have begun to incorporate data geometry into the theoretical analyses of DNNs by applying the assumption that the data lies on a general manifold. First we note the works looking at DNNs from the lens of expressive power combined with the idea of data geometry. Shaham et al. (2015) demonstrate that the size of the neural network depends on the curvature of the data

manifold and the complexity of the function, whilst depending weakly on the input data dimension, for their construction of sparsely-connected 4-layer neural networks. Cloninger & Klock (2020) show that their construction of deep ReLU nets achieve near optimal approximation rates which depend only on the intrinsic dimensionality of the data. Chen et al. (2019) exploit the low dimensional structure of data to enhance the function approximation capacity of Deep ReLU networks by means of theoretical guarantees. Schmidt-Hieber (2019) shows that sparsely connected deep ReLU networks can approximate a Holder function on a low dimensional manifold embedded in a high dimensional space. Simultaneously, researchers have incorporated data geometry into the learning dynamics line of work (Goldt et al., 2020; Paccolat et al., 2020; Buchanan et al., 2021; Wang et al., 2021). Buchanan et al. (2021) apply the NTK model to study how DNNs can separate two curves, representing the data manifolds of two separate classes, on the unit sphere. Goldt et al. (2020) introduce the Hidden Manifold Model for structured data sets to capture the dynamics of two-layer neural networks trained with stochastic gradient descent. Finally, we also note that Rahaman et al. (2019) provide empirical results on which data manifolds are learned faster.

Our work fits into the study of expressive power of DNNs. The number of linear regions is a good proxy for the *practical* expressive power or approximation capacity of Deep ReLU networks (Montúfar et al., 2014). The results surrounding the density of linear regions make the fewest simplifying assumptions both on the data and the architecture of the DNN. The results by Hanin & Rolnick (2019a) bound the number of linear regions orders of magnitude tighter than previous results by deriving bounds for the average case and not the worst case. Moreover, they demonstrate the validity empirically in a setting with very few simplifying assumptions. We introduce the manifold hypothesis to this setting in order to obtain tighter bounds for the first time. This introduces a toolbox of ideas from differential geometry to analyse the approximation capacity of deep ReLU networks.

In addition to the theoretical works listed above, there has been significant empirical work that applies DNNs to non-Euclidean data (Bronstein et al., 2017; 2021). Here the data is assumed to be sampled from manifolds with certain geometric properties. For example, Ganea et al. (2018) design DNNs for data sampled from Hyperbolic spaces of arbitrary dimensionality and modify the forward and backward passes accordingly. There have been numerous applications of modified DNNs, namely graph convolutional networks, to graph data that incorporate the idea that graphs are discrete samples from a smooth manifold (Henaff et al., 2015; Monti et al., 2017; Kipf & Welling, 2017), see Wu et al. (2019) for a comprehensive review. Graph convolutional networks have also been applied to point cloud data for applications in graphics (Qi et al., 2017; Wang et al., 2019).

## D PROOF SKETCH

In this section we provide an overview of how the three main theorems are proved. Theorem 1 provides an equality for measuring the volume of $m - k$ dimensional boundary regions on the manifold. To this effect, we introduce the idea of viewing boundary regions as submanifolds on the data manifold instead of hyperplanes (Proposition 6). We then prove an equality between the volume of boundary regions and the Jacobian of the neurons over the manifold. We utilise the smooth coarea formula that, intuitively, is applied to integrate a function using level sets on a manifold. This completes the proof for Theorem 1.

To prove Theorem 2 we first prove that the Jacobian of a function on a manifold can be denoted using the volume of paralellepiped of vectors in the ambient space subject to a linear transform (Proposition 8). Using this result and combining it with Theorem 1 we can then give an inequality for the density of linear regions. As can be expected this volume depends on the aforementioned projection, which in turn is related to the geometry of the manifold.

Finally, for proving Theorem 3 we first provide an inequality over the tubular neighbourhood of the boundary region. We then use this result to lower bound the geodesic distance between the boundary region and any random point on the manifold. The proof strategy follows that of Hanin & Rolnick (2019a) but there are major deviations when it comes to accounting for the geometry of the data manifold. To the best of our knowledge, we are utilising elements of differential topology that are unique to machine learning when it comes to developing a theoretical understanding of DNNs.

# E  PROOF OF THEOREM 1

We follow the proof strategy used by Hanin & Rolnick (2019a) but deviate from it to account for our setting where $x \in M$. Let $S_z$ be the set of values at which the neuron $z$ has a discontinuity in the differential of its output (or the neuron switches between the two linear regions of the piecewise linear activation $\sigma$),

$$S_z := \{x \in \mathbb{R}^{n_{\text{in}}} | z(x) - b_z = 0\}.$$

We also have

$$\mathcal{O} := \left\{x \in \mathbb{R}^{n_{\text{in}}} | \forall j = 1, ..., L \; \exists \text{ neuron } z \text{ with } l(z) = j \text{ s.t. } \sigma'(z(x) - b_z) \neq 0\right\}.$$

Further,

$$\widetilde{S_z} := S_z \cap \mathcal{O}.$$

We state propositions 9 and 10 by Hanin & Rolnick (2019a) as we apply them to prove Theorem 1, relabeling them as needed.

**Proposition 4.** *(Proposition 9 by Hanin & Rolnick (2019a)) Under assumptions A1 and A2, we have, with probability 1,*

$$B_F = \bigcup_{\text{neurons } z} \widetilde{S_z}.$$

By extending the notion of $S_z$ to multiple neurons we have

$$\widetilde{S}_{z_1,...,z_k} := \bigcap_{j=1}^{k} \widetilde{S}_{z_j},$$

meaning that the set $\widetilde{S}_{z_1,...,z_k}$ is, intuitively, the collection of inputs in $\mathbb{R}^{\text{in}}$ where the neurons $z_j, j = 1, ..., k$, switch between linear regions for $\sigma$ and at which the output of $F$ is affected by the outputs of these neurons. We refer the reader to section B of the appendix in the work by Hanin & Rolnick (2019a) for an intuitive explanation of proposition 4. Next we state proposition 10 by Hanin & Rolnick (2019a).

**Proposition 5.** *(Prosposition 10 by Hanin & Rolnick (2019a)) Fix $k = 1, ..., n_{in}$, and $k$ distinct neurons $z_1, ..., z_k$ in $F$. Then, with probability 1, for every $x \in B_{F,k}$ there exists a neighbourhood in which $B_{F,k}$ coincides with a $n_{in-k}-$dimensional hyperplane.*

We now present Proposition 6, and its proof, which incorporates the additional constraint that $x \in M$, which is an $m$-dimensional manifold in $\mathbb{R}^{n_{\text{in}}}$. To prove the proposition we need the definition of tranversal intersection of two manifolds (Guillemin & Pollack, 1974).

**Definition E.1.** *Two submanifolds, $M_1$ and $M_2$, of $S$ are said to intersect transversally if at every point of intersection their tangent spaces, at that point, together generate the tangent space of the manifold, $S$, by means of linear combinations. Formally, for all $x \in M_1 \cap M_2$*

$$T_x S = T_x M_1 + T_x M_2,$$

*if and only if $M_1$ and $M_2$ intersect transversally.*

For example, given a 2D hyperplane, $P$, and the surface of a 3D sphere, $S^2$, intersect in the ambient space $\mathbb{R}^3$. We have that this intersection is transverse if and only if $P$ is not tangent to $S_2$. For the case where a 2D hyperplane, $\bar{P}$, intersects with $S^2$ at a point $p$ but does not intersect tranversally it coincides exactly with the tangent plane of $S^2$ at point $\{p\} = S^2 \cap P$, i.e. $T_p S = P$. Note that in either case the tangent space of the 2D hyperplane $P$ at any point of intersection is the plane itself.

**Proposition 6.** *Fix $k = 1, ..., m$ and $k$ distinct neurons $z_1, ..., z_k$ in $F$. Then, with probability 1, for every $x \in B_{F,k} \cap M$ there exists a neighbourhood in which $B_{F,k}$ coincides with an $m - k$ dimensional submanifold in $\mathbb{R}^{in}$.*

*Proof.* From Proposition 5 we already know that $B_{F,k}$ is a $n_{\text{in}} - k$-dimensional hyperplane in some neighbourhood of $x$, with probability 1, for any $x \in B_{F,k} \cap M$. Let this hyperplane be denoted by $P_k$. This is an $n - k$ dimensional submanifold of $\mathbb{R}^{n_{\text{in}}}$. The tangent space of this hyperplane at $x$ is the hyperplane itself. Therefore, from assumptions A1 and A2 we have that the probability that this hyperplane intersects the manifold $M$ transversally with probability 1. In other words the probability that this plane $P_k$ contains or is contained in $T_x M$ is 0. Finally, we have the intersection, $M \cap H_k$, has dimension $\dim(M) + \dim(H_k) - n_{\text{in}}$ (Guillemin & Pollack, 1974), which is equal to $m - k$. $\square$

One implication of Proposition 6 is that for any $k \leq m$ the $m - (k+1)$ dimensional volume of $B_{F,k} \cap M$ is 0. In addition to that, Proposition 6 implies that, with probability 1,

$$\text{vol}_{m-k}(\mathcal{B}_{F,k}) = \sum_{\text{distinct neurons } z_1, ..., z_k} \text{vol}_{m-k}(\widetilde{S}_{z_1,...,z_k} \cap M). \tag{3}$$

The final step in the proof of Theorem 1 is to prove the following result.

**Proposition 7.** *Let $z_1, ..., z_k$ be distinct neurons in $F$ and $k \leq m$. Then for a bounded $m-$Hausdorff measurable manifold $M$ embedded in $\mathbb{R}^{n_{in}}$,*

$$\mathbb{E}\left[vol_{m-k}\left(\widetilde{S}_{z_1,...,z_k} \cap M\right)\right] = \int_M \mathbb{E}\left[Y_{z_1,...,z_k}(x)\right]dx,$$

*where $Y_{z_1,...,z_k}(x)$ equals*

$$J_{m,H_k}^M(x)\rho_{b_1,...,b_k}(z_1(x), ..., z_k(x)),$$

*times the indicator function of the event that $z_j$, for $j = 1, ..., k$, is good at $x$ for every $j$ and $H_k : \mathbb{R}^{n_{in}} \to \mathbb{R}^k$ is such that $H_k(x) = [z_1(x), ..., z_k(x)]^T$. The expectation is over the distribution of weights and biases.*

*Proof.* Let $z_1, ..., z_k$ be distinct neurons in $F$ and $M$ be an $m-$dimensional compact Haudorff measurable manifold. We seek to compute the mean of $\text{vol}_{m-k}(\widetilde{S}_{z_1,...,z_k} \cap M)$ over the distribution of weights and biases. We can rewrite this expression as

$$\int_{S_{z_1,...,z_k} \cap M} \mathbf{1}_{z_j \text{ is good at } x} d\text{vol}_{m-k}(x). \tag{4}$$

The map $H_k$ is Lipschitz and $C^1$ almost everywhere. We first note the smooth coarea formula (theorem 5.3.9 by Krantz & Parks (2008)) in context of our notation. Suppose $m \geq k$ and $H_k : \mathbb{R}^{n_{in}} \to \mathbb{R}^k$ is $C^1$ and $M \subseteq \mathbb{R}^{n_{in}}$ is an $m-$dimensional $C^1$ manifold in $\mathbb{R}^{n_{in}}$, then

$$\int_M g(x)J_{k,H_k}^M(x)d\text{vol}_m(x) = \int_{\mathbb{R}^k}\int_{M \cap H_k^{-1}(y)} g(y)d\text{vol}_{m-k}(y)d\text{vol}_k(x), \tag{5}$$

for every $\mathcal{H}^m$-measurable function $g$ where $J_{k,H_k}^M$ is as defined in Definition 3.1.

We denote preactivations and biases of neurons as $\mathbf{z}(x) = [z_1(x), ..., z_k(x)]^T$ and $\mathbf{b_z} = [b_{z_1}, ..., b_{z_k}]^T$. From the notation in A1, we have that

$$\rho_{\mathbf{b_z}} = \rho_{b_{z_1},...,b_{z_k}},$$

is the joint conditional density of $b_{z_1}, ..., b_{z_k}$ given all other weights and biases. The mean of the term in equation 4 over the conditional distribution of $b_{z_1}, ..., b_{z_k}$, $\rho_{\mathbf{b_z}}$, is therefore

$$\int_{\mathbb{R}^k} \mathbf{b}d\text{vol}_k(\mathbf{b}) \int_{\{\mathbf{z}=\mathbf{b}\} \cap M} \mathbf{1}_{z_j \text{ is good at } x} d\text{vol}_{m-k}(x), \tag{6}$$

where we denote $[b_1, ..., b_k]^T$ as $\mathbf{b}$. Thus applying the smooth co-area formula (Equation 5) to the expression in 6 shows that the average 4 is equal to

$$\int_M Y_{z_1,...,z_k}(x)dx.$$

Finally, we take the average over the remaining weights and biases and commute the expectation with the $dx$ integral. We can do this since the integrand is non-negative. This gives us the result:

$$\mathbb{E}\left[\text{vol}_{m-k}\left(\widetilde{S}_{z_1,...,z_k} \cap M\right)\right] = \int_M \mathbb{E}\left[Y_{z_1,...,z_k}(x)\right]dx, \tag{7}$$

as required. $\square$

Finally, taking the summation over all possible sets of distinct neurons $z_1, ..., z_k$ and combining equation 3 with Proposition 7 completes the proof for Theorem 1.

## F  PROOF OF THEOREM 2

To prove the upper bound in Theorem 2 we first show that the (determinant of) Jacobian for the function $H_k : M \to \mathbb{R}^k$, $H_k(x) = [z_1(x), ..., z_k(x)]^T$, as defined in 3.1 is equal to the volume of the parallelopiped defined by the vectors $\phi_{H_k}(\nabla z_j(x))$, for $j = 1, ..., k$, where $\phi_{H_k} : \mathbb{R}^k \to T_x M$ is an orthogonal projection onto the orthogonal complement of the kernel of the differential $D_M H_k$. Intuitively, this shows that with the added assumption $x \in M$ in Theorem 2 how exactly we can incorporate the geometry of the data manifold $M$ into the upper bound provided by Hanin & Rolnick (2019a) in corollary 7.

**Proposition 8.** *Given $H_k : M \to \mathbb{R}^k$ such that $H_k(x) = [z_1(x), ..., z_k(x)]^T$ and the differential $D_M H_k$ is surjective at $x$ then*

$$J^M_{k,H_k}(x) = \sqrt{\det(Gram(\phi_{H_k}(\nabla z_1(x)), ..., \phi_{H_k}(\nabla z_k(x))))}, \tag{8}$$

*where $\phi_{H_k} : \mathbb{R}^n \to \mathbb{R}^k$ is a linear map and Gram denotes the Gramian matrix.*

*Proof.* We first define the orthogonal complement of the kernel of the differential $D_M H_k$. For a manifold $M \subset \mathbb{R}^n$ and a fixed point $x$ we have that $T_x M$ is a $m-$dimensional hyperplane. If we choose an orthonormal basis $e_1, ..., e_n$ of $\mathbb{R}^n$ such that $e_1, ..., e_m$ spans $T_x M$ for a fixed $x$ we can denote all vectors in $T_x M$ using $m$ coordinates corresponding to this basis. Therefore, for any vector $y \in \mathbb{R}^k$ we can get the orthogonal projection of $y$ onto $T_x M$ using a $m \times n$ matrix which we denote as $P_x$, where $P_x y$ (matrix multiplied by a vector) represents a vector in $T_x M$ corresponding to the basis $e_1, ..., e_m$. For any manifold $M$ in $R^n$ and function $H_k : M \to \mathbb{R}^k$ we have that $D_M H_k : T_x M \to \mathbb{R}^k$ at a fixed point $x$ is linear function. Therefore we can write $D_M H_k(v) = Av$ where $v \in T_x M$ is denoted using the aforementioned basis of $T_x M$. This implies that $A$ is a $k \times m$ matrix. Therefore, the kernel of $D_M H_k$ for a fixed point $x \in M$ is

$$\ker(D_M H_k) = \Big\{ z | Az = 0 \text{ and } z \in T_x M \Big\}.$$

Since we can create a canonical basis for the space $\ker(D_M H_k)$ starting from the basis $e_1, ..., e_m$ in $R^n$ using the Gram-Schmidt process given the matrix $A$ we have that for any $y \in R^n$ we can project it orthogonally onto $\ker(D_M H_k)$. The orthogonal complement of $\ker(D_M H_k)$ is therefore defined by

$$\ker(D_M H_k)^{\perp} = \Big\{ a | a \cdot z = 0 \text{ for some } z \in \ker(D_M H_k) \text{ and } a \in T_x M \Big\}.$$

Similar to the previous argument, we construct a canonical basis starting from $e_1, ..., e_m$ for $\ker(D_M H_k)^{\perp}$ and therefore we can denote the orthogonal projection onto $\ker(D_M H_k)^{\perp}$ as a linear transformation. We denote this linear projection for fixed $x$ using $\phi_k$.

We denote the basis vectors $e_1, ...., e_m$ as a $m \times n$ matrix $E$ where each row $i$ corresponds to the vector $e_i$. Therefore, the orthogonal projection of any vector $y \in \mathbb{R}^n$ is $Ey$. Now we can get the matrix $A$ using $E\nabla z_j(x)$ corresponding to each row $j$ for $j = 1, ..., m$. This uses the fact that the direction of steepest ascent on $z_j(x)$ restricted to the tangent space $T_x M$ of the manifold $M$ is an orthogonal projection of the direction of steepest ascent in $\mathbb{R}^n$.

Finally, from lemma 5.3.5 by Guillemin & Pollack (1974) we have that

$$J^M_{k,H_k}(x) = \mathcal{H}^k(D_M H_k(P))/\mathcal{H}^k(P),$$

for any parallelepiped $P$ contained in $(\ker(D_M H_k))^{\perp}$. Arguing similar to the proof of lemma 5.3.5 by Guillemin & Pollack (1974) we get that

$$J^M_{k,H_k}(x) = \sqrt{\det((A)^T A)} = \sqrt{\det Gram(E\nabla z_1(x), ..., E\nabla z_k(x))},$$

thereby showing that $\phi_{H_k}(y) = Ey$ is a linear mapping.  □

Although we state Proposition 8 for neurons $z_j(x), j = 1, ..., k$ in the proof, it applies to any function that satisfy the conditions laid out in the proposition. Equipped with Proposition 8 we prove Theorem

2. When the weights and biases of $F$ are independent obtain an upper bound on $\rho_{b_{z_1},...,b_{z_k}}(b_1,...,b_k)$ as

$$\Pi_{j=1}^k \rho_{b_{z_j}}(b_1,...,b_k) \leq \left( \sup_{\text{neurons } z} \rho_{b_z}(b) \right)^k = C_{\text{bias}}^k.$$

Hence,

$$Y_{z_1,...,z_k} \leq C_{\text{bias}}^k J_{k,H_k}^M.$$

From Proposition 8 we have that $J_{k,H_k}^M$ is equal to the $k$-dimensional volume of the paralellopiped spanned by $\phi_x(\nabla z_j(x))$ for $j = 1,...,k$. Therefore, we have

$$J_{k,H_k}^M \leq \Pi_{j=1}^k ||E\nabla z_j(x)|| \leq ||E||^k \Pi_{j=1}^k ||\nabla z_j(x)||, \tag{9}$$

where $||E||$ denotes the matrix norm which is defined as

$$||E|| = \sup\left\{ ||Ey|| \Big| y \in \mathbb{R}^k, ||y|| = 1 \right\}.$$

Note that $E$ does not depend on $F$ (or $z_1,...,z_k$) but only on $T_xM$ or more generally the geometry of $M$ at any point $x$. From theorem 1 by Hanin & Nica (2018) we have, for any fixed $x$,

$$\mathbb{E}\left[ \Pi_{j=1}^k ||\nabla z_j(x)|| \right] \leq \left( C_{\text{grad}} \right)^k, \tag{10}$$

where,

$$C_{\text{grad}} = \sup_z \sup_{x \in \mathbb{R}^{n_{\text{in}}}} \mathbb{E}[||\nabla z(x)||^{2k}]^{1/k} \leq Ce^{C\sum_{j=1}^d \frac{1}{n_j}},$$

wherein $C > 0$ depends only on $\mu$ and not on the architecture of $F$ and $n_j$ is the width of the hidden layer $j$. Let $C_M$ be defined as

$$C_M := \sup \Big\{ C | \text{ there exists a set, S, of non zero } m - k\text{-dimensional Hausdorff measure}$$

$$\text{such that } ||E_x|| \geq C \forall x \in S \Big\}$$

Therefore, combining equations 10, 9 and result from Theorem 1 we have

$$\frac{\mathbb{E}[\text{vol}_{m-k}(\mathcal{B}_{F,k} \cap M)]}{\text{vol}_m(M)} \leq \binom{\text{number of neurons}}{k} (2C_{\text{grad}} C_{\text{bias}} C_M)^k,$$

where the expectation is over the distribution of weights and biases.

## G    PROOF OF THEOREM 3

We first prove the following proposition

**Proposition 9.** *For a compact $m$-dimensional submanifold $M$ in $\mathbb{R}^n$, $m, n \geq 1$ and $m < n$ let $S \subseteq \mathbb{R}^n$ be a compact fixed continuous piecewise linear submanifold with finitely many pieces and given any $U > 0$. Let $S_0 = \emptyset$ and let $S_k$ be the union of the interiors of all $k$-dimensional pieces of $S \setminus (S_0 \cup ... \cup S_{k-1})$. Denote by $T_\epsilon$ the $\epsilon$-tubuluar neighbourhood of any $X \subset M$ such that*

$$T_\epsilon(X) = \Big\{ y | d_M(y, X) < \epsilon \text{ and } y \in M \Big\},$$

*where $\epsilon \in (0, U)$, $d_M$ is the geodesic distance between the point $y$ and set $X$ on the manifold $M$, we have*

$$\text{vol}_m(T_\epsilon(S)) \leq \sum_{k=n-m}^d \text{vol}_k(S_k \cap M)\omega_{n-k}\epsilon^{n-k} C_{k,\kappa,U},$$

*where $C_{k,\kappa,U} > 0$ is a constant that depends on the average scalar curvature $\kappa_{(S_k \cap M)^\perp}$ and $U$, and $\omega_{n-k}$ is the volume of the unit ball in $\mathbb{R}^{n-k}$.*

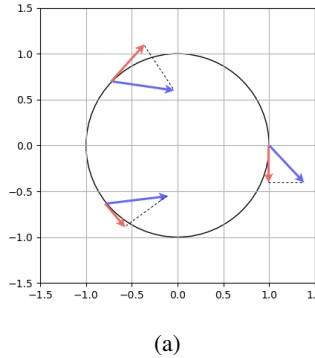
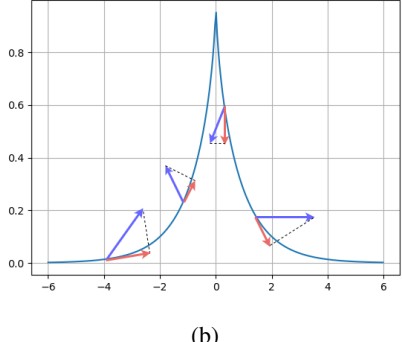

(a)                                     (b)

Figure 11: We illustrate how vectors project differently on tangent planes of two different manifolds: circle (a) and tractrix (b). In case of the tractrix the tangents (and the projection of vectors onto them) are on the inside of the tractrix whereas for the sphere the tangents are always on the outside of the sphere. Since the projections of vectors onto the tangent space are an essential aspect of our proof we end up with the term $C_M$, which quantifies the "shrinking" of these vectors upon projection, in the inequalities for Theorems 2 and 3

*Proof.* Define $d$ to be the maximal dimension of linear pieces in $S$. Let $x \in T_\epsilon(X \cap M)$. Suppose $x \notin T_\epsilon(X \cap M)$ for all $k = n - m, ..., d - 1$. Then the intersection of a geodesic ball of radius $\epsilon$ around $s$ with $S$ is a ball inside $S_d \cap M$. Using the convexity of this ball, with respect to the manifold $M$ (Robbin et al., 2011), there exists a point $y$ in $S_d \cap M$ such that the geodesic $\gamma : [0, 1] \to M$ with $\gamma(0) = y$ and $\gamma(1) = x$ is perpendicular to $S_d \cap M$ at $y$. Formally, $T_{S_d \cap M} M$ at $y$ is perpendicular to $\dot{\gamma}(0) \in T_M$ at $y$. Let $B_\epsilon(N^*(S_d \cap M))$ be the union of all the $\epsilon$ balls along the fiber of the submanifold $S_d \cap M$. Therefore, we have

$$\text{vol}_m(T_\epsilon(S \cap M) \leq \text{vol}_m(B_\epsilon(N^*(S_d \cap M)) + \text{vol}_m(T_\epsilon(S_{\leq d-1} \cap M)), \tag{11}$$

where $S_{\leq d-1} := \cup_{k=0}^{d-1} S_k$. We also note that

$$\text{vol}_m(B_\epsilon(N^*(S_d \cap M))) = \text{vol}_{m+d-n}(S_d \cap M)\text{vol}_{n-d}(B_\epsilon((M \cap S_d)^\perp)),$$

where $B_\epsilon((M \cap S_d)^\perp)$ is the average volume of an $\epsilon$ ball in the submanifold of $M$ orthogonal to $M \cap S_d$. This volume depends on the average scalar curvature, $\kappa_{(M \cap S_d)^\perp}$ of the submanifold $(M \cap S_d)^\perp$. As shown by Wan (2016), for a fixed point $x \in (M \cap S_d)^\perp$

$$\text{vol}_{n-d}(B_\epsilon(x, (M \cap S_d)^\perp)) = \omega_{n-d}\epsilon^{n-d}\Big(1 - \frac{\kappa(x)_{(M \cap S_d)^\perp}}{n - d + 2}\epsilon^2 + O(\epsilon^4)\Big),$$

where $\omega_{n-d}$ is the volume of the unit ball of dimension $n - d$, $B_\epsilon(x, (M \cap S_d)^\perp)$ is the geodesic ball of radius $\epsilon$ in the manifold $(M \cap S_d)^\perp$ centered at $x$ and $\kappa_{(M \cap S_d)^\perp}(x)$ denotes the scalar curvature at point $x$. Gray (1974) provides the second order expansion of the formula above. Given that $\epsilon \in (0, U)$, for all $k \in \{n - m, n - m + 1, ..., d\}$, then we have a smallest $C_{k,\kappa,U}$ such that

$$\text{vol}_k(B_\epsilon(x, (M \cap S_k)^\perp)) \leq C_{k,\kappa,U}\epsilon^k. \tag{12}$$

The above inequality follows from assumption A5. Using the above inequalities 11, 12 and repeating the argument $d - 1 - n + m$ times we get the result of the proposition. $\square$

We also note that $C_{k,\kappa,U}$ increases monotonically with $U$, this also follows from the volume being monotonically increasing and positive for $\epsilon > 0$. Finally, we can now prove Theorem 3. Let $x \in M$

be uniformly chosen. Then, for all $\epsilon \in (0, U)$, using Markov's inequality and Proposition 9, we have

$$
\begin{aligned}
\mathbb{E}[\text{distance}_M(x, B_f \cap M)] &\geq \epsilon \Pr(\text{distance}_M(x, B_F \cap M) > \epsilon) \\
&= \epsilon(1 - \Pr(\text{distance}_M(x, B_F \cap M) <= \epsilon)) \\
&\geq \epsilon\left(1 - \sum_{k=n_{\text{in}}-m}^{n_{\text{in}}} \text{vol}_k(S_k \cap M)\omega_{n-k}\epsilon^{n_{\text{in}}-k}C_{n_{\text{in}}-k,\kappa,U}\right) \\
&\geq \epsilon\left(1 - \sum_{k=n_{\text{in}}-m}^{n_{\text{in}}} C_{n_{\text{in}}-k,\kappa,U}(C_{\text{grad}}C_{\text{bias}}C_M\epsilon\{\#\text{neurons}\})^k\right).
\end{aligned}
$$

Note that as we increase $U$ the constants $C_{n-k,\kappa,U}$ increase, although not strictly, for all $k$. To find the supremum of the expression on the right hand side, of the last inequality, in $\epsilon \in (0, U)$ we multiply and divide the expression by $C_{\text{grad}}C_{\text{bias}}C_M\#\text{neurons}$ to get the polynomial

$$
p_U(\zeta) = \frac{\zeta\left(1 - \sum_{k=n_{\text{in}}-m}^{n_{\text{in}}} C_{n_{\text{in}}-k,\kappa,U}\zeta^k\right)}{C_{\text{grad}}C_{\text{bias}}C_M\#\text{neurons}},
$$

where $\zeta = \epsilon C_{\text{grad}}C_{\text{bias}}C_M\#\text{neurons}$ and $\zeta \in (0, U')$ where $U' = UC_{\text{grad}}C_{\text{bias}}C_M\#\text{neurons}$. Let $d_M$ be the diameter of the manifold $M$, defined by $d_M = \sup_{x,y\in M} \text{distance}_M(x, y)$. We assume that $d_M$ is finite. Taking the supremum over all $U \in (0, d_M]$ or $U' \in (0, d'_M]$, where $d'_M = d_M C_{\text{grad}}C_{\text{bias}}C_M\#\text{neurons}$, gives us the constant $C_{M,\kappa}$

$$
C_{M,\kappa} = \sup_{U'\in(0,d'_M]} \{ \sup_{\zeta\in(0,U')} \{p_U(\zeta)\}\}.
$$

Since $d_M$ is finite the constant above exists and is finite. We make a note on the existence of this constant $C_{M,\kappa}$ in the absence of the constraint that the diameter of manifold $M$ is finite. As $U$ increases the constants $C_{n_{\text{in}}-k,\kappa,U}$ also increase and are all positive. The solution for $p'_U(\zeta) = 0, \zeta > 0$, which we denote by $\zeta_U$, is unique and keeps decreasing as $U$ increases. The uniqueness of the solution follows from the fact that the coefficients $C_{n_{\text{in}}-k,\kappa,U}$ are all positive. We also note that $p_U(\zeta_U)$ need not be equal to $\sup_{\zeta\in(0,U')}\{p_U(\zeta)\}$ because $\zeta_U$ need not lie in $(0, U')$. In all such cases $\sup_{\zeta\in(0,U')}\{p_U(\zeta)\} = p_U(U')$. Given the polynomial $p_U(\zeta)$ above if we can assert that there exists a $C_U$, and the corresponding $C_{U'}$, such that for all $U > C_U$, and corresponding $U' > C_{U'}$, we have $\sup_{\zeta\in(0,U')}\{p_U(\zeta)\} = p_U(\zeta_U) < \infty$ and for all $0 < U \leq C_U$ we have $\sup_{\zeta\in(0,U')}\{p_U(\zeta)\} = p_U(U') < \infty$. Therefore, $C_{M,\kappa}$ exists and is finite if the previous assertion holds, proving this assertion is beyond the scope of our current work and particularly challenging.

Finally, taking the average over distribution of weights gives us the inequality

$$
\mathbb{E}[\text{distance}_M(x, B_f \cap M)] \geq \frac{C_{M,\kappa}}{C_{\text{grad}}C_{\text{bias}}C_M\#\text{neurons}},
$$

where $C_{M,\kappa}$ is a constant which depends on the average scalar curvature of the manifold $M$. This completes the proof of Theorem 3.

## H   TOY SUPERVISED LEARNING PROBLEMS

For the two supervised learning tasks with different geometries (tractrix and sphere), we uniformly sample 1000 data points from each 1D manifold to come up with samples of $(x_i, y_i)$ pairs. We then add Gaussian noise to $y$. We train a DNN with 2 hidden layers, with 10 and 16 neurons in each layer and a single linear output neuron, for a total of 26 neurons with piecewise linearity, using the PyTorch library. The optimization is performed using the Adam optimizer (Kingma & Ba, 2015) with a learning rate of 0.01. We ensure a reasonable fit of the model by reducing the test time mean squared error (see Figure 12). We then calculate the exact number of linear regions on the respective domains by finding the points where $z(x) = b_z$ for every neuron $z$ and $x$ is on the 1D manifold. We do this by adding neurons, $z$, one by one at every layer and using the SLSQP (Kraft, 1988) to solve for $|z(x) - b_z| = 0$. We then split a linear region depending on where this solution lies compared to previous layers. For every epoch, we then uniformly randomly sample points from the 1D manifold,

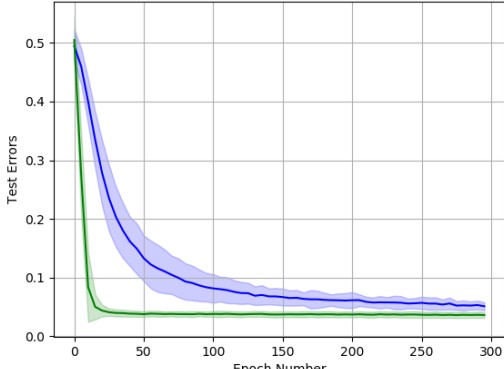

Figure 12: The test errors for the cases where data is sampled from the tractrix (blue) and the circle (green). We see that the tractrix converges slower but the magnitude of the errors remains comparable as training progresses across the two manifolds.

by sampling directly from $\theta$ and $t$, to measure average distance to the nearest linear boundaries. The experiment was run on CPUs, from training to counting of number of linear regions. The intel cpus had access to 4 GB memory per core. A total of, approximately, 24 cpu hours were required for all the experiments in this section. This was run on an on demand cloud instance. All implementations are in PyTorch, except for SLQSP for which we used sklearn.

## H.1 VARYING $n_{\text{IN}}$

The experimental setup, hyperparameters, network architecture, target function and methods are all the same as described for the toy supervised learning problem for the case where the geometry is a sphere. The only difference is that the input dimension varies, $n_{\text{in}}$.

## I HIGH DIMENSIONAL DATASET

We utilise the official implementation of pretrained StyleGAN generator to generate curves of images that lie on the manifold of face images. Specifically, for each curve we sample a random pair of latent vectors: $z_1, z_2 \in \mathbb{R}^k$, this gives us the start and end point of the curve using the generator $g(z_1)$ and $g(z_2)$. We then generate 100 images to approximate a curve connecting the two images on the image manifold in a piece-wise manner. We do so by taking 100 points on the line connecting $z_1$ and $z_2$ in the latent space that are evenly spaced and generate an image from each one of them. Therefore, the $i^{\text{th}}$ image is generated as: $x_i = g(((100-i) \times z_1 + i \times z_2)/100)$, using the StyleGAN generator $g$. We qualitatively verify the images to ensure that they lie on the manifold of images of faces. 4 examples of these curves, sampled as above, are illustrated in the video here: `https://drive.google.com/file/d/1p9B8ATVQGQYoiMh3Q22D-jSaI0USsoNx/view?usp=sharing`.

The neural network, used for classification in our MetFaces experiment, is feed forward with ReLU activation. There are two hidden layers with 256 and 64 neurons in the first and second layers respectively. We downsample the images to $128 \times 128 \times 3$. We augment the dataset using random horizontal flips of the images. All inputs are normalized. We use a batch size of 32. The neural network is trained using SGD. The learning rate is 0.01 and the momentum is 0.5. The total time required, for these experiments on metfaces dataset, was approximately 36 GPU hours on a Titan RTX GPU that has 24 GB memory. This was run on an on demand cloud instance. We chose hyperparameters by trial and error, targeting a better fit for the training data.

## J    CODE, DATA AND LICENSES

All the code used for our experiments (except the StyleGAN2 code) is enclosed in the folder `exp/`. The instructions to run all the experiments are enclosed in `exp/readme.txt`. We plan on releasing the code as an open github repository under the MIT License (`https://opensource.org/licenses/MIT`). The files changed on the github repository for the official implementation of StyleGAN2 (`https://github.com/NVlabs/stylegan2-ada-pytorch`) are enclosed in the folder `stylegan2-ada-pytorch`. The instructions to run the experiments are documented in `stylegan2-ada-pytorch/readme.txt`.

Finally, the images we used to sample linear regions on a curve's piecewise approximation on the manifold of face images, for the MetFaces experiment, are in the zip file `https://drive.google.com/file/d/1x5t-sc9NlW5N_ZBXUM0WcfX-toUXa85L/view?usp=sharing`.

## K    BROADER IMPACT STATEMENT

Although DNNs have been highly effective in approximating arbitrarily complex functions the reason behind their effectiveness remains open. Similarly, the cases and theoretical reasons for where they fail to approximate or introduce bias remain underexplored. Our work is unique in the sense that it looks at both data and model together when estimating the approximation capabilities of the model. Our empirical results are also close to the theoretical predictions which enables us to be more confident of our assertions. The negative impact and ethical concern that our work has, and it pertains to deep learning as a whole, is that we use a lot of compute. This has two dimensions of concern: **1)** the environmental costs are fairly high, and **2)** further research in deep learning that uses excessive compute creates barriers for smaller labs and individuals from training state of the art models.

