# OpenReview forum: "Effects of Data Geometry in Early Deep Learning"
_ICLR.cc/2022/Conference — ICLR 2022 Submitted_

### Official Review · Reviewer_WSLJ · 2021-11-02

**Correctness:** 3
**Technical Novelty And Significance:** 3
**Empirical Novelty And Significance:** 2
**Recommendation:** 5
**Confidence:** 4

**Main Review:**

Strengths:

The structure of the paper is clear. I believe that the study of linear regions of ReLU networks possesses value and has interesting implications. Taking data intrinsic structures into consideration is very important in understanding deep learning, as pointed out by the authors, practical data are often low-dimensional due to local regularities, symmetries, etc. In addition to theoretical study, the paper also provides numerical experiments, in support of the theory.

Weakness:

1. Technical ambiguity. The presentation of theoretical results suffer from some ambiguity. For example, on page 5 before Definition 3.1, the explanation of Jacobian of a function may be elaborated with formal mathematical expressions and necessary verbal description. Another example is in Theorem 1. Quantity $Y_{z_1, \dots, z_k}$ is defined by some expression, times the indicator function of the event that $z_j$ is good at $x$ for $j = 1, \dots, k$. The reviewer found this paragraph very complicated and hard to follow. It can be better explained to define $Y_{z_1, \dots, z_k}$ before the statement of the theorem.

Theorem 2 and 3 both involve several parameters $C_{\rm xxx}$ depending on the geometry of manifold, initialization of weights, etc. However, even looking into the proofs, these parameters are not explicitly given. This in turn makes clear interpretations of Theorem 2 and 3 elusive. The reason I am asking this is two-fold: (1) on page 4, the results in Hanin & Rolnick (2019a) are summarized in a very clean form. Nonetheless, the results in Theorem 2 and 3 seem not comparable with Hanin & Rolnick (2019a). Some quick glance can result in with data intrinsic structures, Theorem 2 indicates a worse polynomial dependence on the size of networks. Moreover, it is possible to choose an optimal $\epsilon$ in Theorem 3. (2) In experiments, there are extensive discussion on $C_M$ and $C_{\rm grad}$. While without any explicit formulas, it is hard to expect how will $C_M$ and $C_{\rm grad}$ scale with the problem setting.

2. Unclear connection with expressivity of neural networks. The experiments devote to plot the number of linear regions resulting from initialization and training. However, there is no connection between the performance of the network with the number of linear regions. For example, in periodic regression problems, what is the testing error? Does it have any correlation with the number of linear regions across 20 runs? When changing the number of neurons, does the number of linear regions change correspondingly as predicted by Theorem 2?





**Summary Of The Paper:**

This paper theoretically characterizes the number of linear regions of ReLU neural networks at initialization. Compared to existing results, the authors take the data manifold into consideration, and the number of linear regions is measured on the data manifold. Moreover, for uniformly random sampled data on a compact manifold, the expected (geodesic) distance to the linear boundary is analyzed.

To further support the theory and also extend to the behavior through the training process, the paper provides synthetic data and real data experiments. There are indications of the number of linear regions does not deviate significantly from their initialization and is faithful to the data manifold dimension.

**Summary Of The Review:**

The paper displays a systematic treatment of studying the number of linear regions in ReLU neural networks. However, the theories and experiments have some issues, which undermine the overall quality of the paper.

---

> ### Author Response · Authors · 2021-11-15
> **Response to Reviewer WSLJ**
>
> Thank you for your insightful comments and the time spent carefully evaluating our work.
>
> We agree that the definition of the Jacobian could use some more explaining. We have added a precise mathematical definition of the objects used in Definition 3.1 in the Appendix and pointed it out in our definition. We have also added an intuitive and textual description of the differential map which plays an important role in the definition. Please take a look and let us know if there are any further clarifications required.
>
> We have indeed provided, in our initial submission, a precise definition of C_M in the Appendix F. We use the constant C_grad from [1] and had cited as such in our initial submission but have now added the precise definition in the Appendix F in the revised version. The goal of our work is to keep the contents approachable whilst making sure all the definitions are precise. In this effort, as rightly noted by the reviewer, we have relegated some of the heavy mathematical definitions to the Appendix whilst providing an intuitive and experimental approach to understanding these constants in the main body of the paper.
>
> On “Unclear connection with expressivity of neural networks”. The connection between density of linear regions as a proxy for expressivity of DNNs is well established [2, 3, 4, 5]. The goal of our work is to study this proxy as it would behave on the data manifold, as it serves as a more practical measure of expressivity. We also note that formalising the connection between density of linear regions and the expressivity of DNNs is beyond the scope of our current work. Having said that, we have added graphs for the test error for both the cases of tractrix and the circle in Appendix H of our revised version (figure 12). We hope this resolves your concern.
>
> When changing the number of neurons, does the number of linear regions change correspondingly as predicted by Theorem 2? -> Showing this is the intention of Figure 8 in the paper.  We apologize for not making that clear.
>
> [1] B. Hanin and M. Nica. Products of many large random matrices and gradients in deep neural networks.
>
> [2] Guido Montúfar, Razvan Pascanu, Kyunghyun Cho, and Yoshua Bengio. On the number of linear regions of deep neural networks.
>
> [3] Telgarsky, M. Representation benefits of deep feedforward networks.
>
> [4] Serra, T., Tjandraatmadja, C., and Ramalingam, S. Bounding and counting linear regions of deep neural networks.
>
> [5] Raghu, M., Poole, B., Kleinberg, J., Ganguli, S., and Dickstein, J. S. On the expressive power of deep neural networks.

---

> > ### Author Response · Authors · 2021-11-24
> > **Follow up on recent draft**
> >
> > We are adding further clarifications after our most recent update to the draft.
> >
> > "​​Moreover, it is possible to choose an optimal epsilon in Theorem 3." -> We have modified the statement of theorem 3 to do just that. The theorem looks similar to that by Hanin and Rolnick in format but incorporates data geometry into the statement and proof. We request you to please take a look.
> >
> > “Theorem 2 indicates a worse polynomial dependence on the size of networks” -> We note that we show an upper bound for the m - k dimensional density of (instead of n - k as done by Hanin and Rolnick). Where m is the dimensionality of the data manifold and n is that of the input data. In form, this is very similar to corollary 4 by Hanin and Rolnick although it incorporates data geometry into the bounds.
> >
> > We hope this addresses all of your concerns and look forward to hearing back from you.

---

### Official Review · Reviewer_5YsV · 2021-11-03

**Correctness:** 4
**Technical Novelty And Significance:** 3
**Empirical Novelty And Significance:** Not applicable
**Recommendation:** 6
**Confidence:** 2

**Main Review:**

The main results extend the recent results by Hanin and Rolnick 19’ on the density of linear regions and the distance from points to the linear boundary from a compact set to the manifold setting. The extension is meaningful in that it takes advantage of the structure of high-dimensional data and makes the theoretical results more applicable to real-world data.

The bounds depend on the number of neurons, neural net architecture, the distribution of biases,  and the geometry of the manifold.  The experiments on two toy datasets were designed to demonstrate the effects of manifold geometry and the neural net architecture. Although the toy example helps understand the effects of all terms. There seems to be a gap between the toy example and applying the analysis to real-world data. If C_M of real datasets are unknown, can the author give examples of what kind of analyses can be done besides the one example in Figure 9. Overall I think the paper can benefit from more empirical results and discussion.


**Summary Of The Paper:**

The paper investigates the number of linear regions of deep neural nets. More specifically how deep neural nets split the input data manifold into regions where it behaves approximately linearly.  It generalizes the results by Hanin and Rolnick 19’ to the manifold setting. Two theoretical bounds: 1) an upper bound for the number of linear regions and 2) a lower bound for the average manifold distance between points on the manifold to the linear boundary. Experiments were done on both toy datasets and MetFaces dataset.

**Summary Of The Review:**

Overall I think the paper can benefit from more empirical results and discussion. While the extension to the manifold setting is a valuable contribution by itself. The significance lacks empirical support.

---

> ### Author Response · Authors · 2021-11-15
> **Response to Reviewer 5YsV**
>
> We thank you for your valuable feedback and confidence in our work. We also thank you for noting that it is a meaningful extension to the work by Hanin and Rolnick.
>
> On further empirical results: we have added proposals for future work from our observations in figure 9. Succinctly, we suggest that "concentrating" linear regions around the data manifold can lead higher expressivity for the same number of parameters. We are supporting our theoretical results with empirical findings wherever feasible: that number of linear regions is indeed different across manifolds (figure 4) and that the number of linear regions scales linearly with number of neurons for 1D manifolds (figure 8). The main message of our work still pertains to the theoretical results.

---

### Official Review · Reviewer_1oFk · 2021-11-06

**Correctness:** 3
**Technical Novelty And Significance:** 2
**Empirical Novelty And Significance:** 2
**Recommendation:** 5
**Confidence:** 4

**Main Review:**


### Strengths

- The study of data with low-dimensional manifold structure is important for
  understanding practical performance of neural networks, so the problem the
  authors study is well-motivated.
- Some of the geometric results that are relied on to enable the proofs are
  somewhat esoteric, at least with respect to the ML literature. Therefore the
  fact that the authors have collected them in their proofs may be a useful
  contribution. Although I have not checked every proof line by line (see some
  comments below), roughly the right kinds of tools seem to be in use here
  (transversality; manifold coarea formula; mean curvature estimates;
  geodesics; ...).

### Weaknesses

- The theoretical content of the paper is very incremental -- in effect, it
  consists of a translation of the results of Hanin and Rolnick to the setting
  of manifold data. In particular, almost all arguments of Hanin and Rolnick
  are reproduced in the proofs of this paper (the theorems are in one-to-one
  correspondence, for a start), just with the corresponding concepts for
  manifolds substituted in (for example, replace Hanin and Rolnick's coarea
  formula with the smooth coarea formula; replace integration over euclidean
  space with integration with respect to the volume form on the manifolds, and
  handle the corresponding curvature and "reach" quantities that arise, which
  leads to the new constants relative to the theorems of Hanin and Rolnick). As
  in the previous section, there may be a useful contribution just in the
  organization of these manifold concepts; on the other hand, the *stylistic*
  similarities between the proofs of the authors and the corresponding proofs
  of Hanin and Rolnick is sometimes uncanny (for example, compare the content
  of section E of Hanin and Rolnick to the content of section F of the current
  submission).
- In a similar vein, the writing style of the authors is often imprecise, in
  that the framework of the paper is so identical to that of Hanin and Rolnick
  that certain notions that were defined in Hanin and Rolnick's paper are taken
  for granted by the authors. For example, the notations for the neural
  network's neurons at the top of page 4 are not defined correctly because
  Hanin and Rolnick do not define these precisely (the weights in the formula
  should be "full", rather than single neurons, and the "column of weights"
  does not make sense here, unless the notation is transposed for some reason
  relative to equation (1)); the discussion of dimensions at the bottom of page
  4 does not mention transversality (it is discussed in the appendix) or in
  particular that these results hold only almost surely; nowhere is it
  mentioned concretely that the weights of $F$ need to satisfy certain
  assumptions on the density until the beginning of the appendix (these are
  important e.g. in the context of the transversality claim made in the body,
  because this is not true for general initializations).
- The presentation of theorem 3 seems to be unnecessarily uninsightful: for
  example, one naturally wants to understand when this result is
  non-vacuous. In addition, the proof has some odd aspects: it seems to
  use an asymptotic statement for $C_{\kappa, k}$ in $\epsilon$, but the
  theorem asserts the result for all $\epsilon > 0$ and does not show that this
  constant depends on $\epsilon$. It seems that this result needs to be
  rewritten to be asymptotic for small $\epsilon$ -- this should also enable a
  more precise conclusion to be asserted, similar to in Hanin and Rolnick
  2019a. This makes me feel like the other proofs may need to be checked for
  minor errors or imprecisions that affect the statements of the results
  (specifically, the constants).
- The experiments conducted and the authors' discussion of them does not
  provide very much insight into the unique circumstances present in the
  manifold case, relative to what is known and expected from Hanin and
  Rolnick's work. For example, the toy experiments are helpful to see the
  effect of curvature, but this should be investigated more systematically
  across many geometries to give a sense of what is going on (the authors only
  provide minimal interpretation of their results in the captions of each
  figure). The StyleGAN results should be discussed more at length -- only one
  figure is presented, and it is not completely clear what the implication is
  (the task consists of fitting a neural network to face images with random
  labels, and then looking at "off manifold" and "on manifold" linear region
  counts, which are different -- what is the implication of this study for
  practice?). This is important because it relates to the authors' stated aim
  for the paper.
- There are missing references to highly related work motivated similarly to
  the authors. The authors only mention other theoretical works on interactions
  between neural networks and manifold data in passing in a sentence in section
  5 -- a more detailed comparison should be made that highlights what the
  authors contribute to this context. In addition, the only works the authors
  reference are about approximation properties of deep neural networks on
  manifolds: there are also very relevant works on algorithmic results for
  training DNNs to perform classification/regression on manifolds, e.g. [2-6]
  below. Again, it would be best to discuss this context in detail in the
  related work/intro.


### Minor Points
- It is best not to use wikipedia links (bottom of page 6) -- these should be
  changed to the corresponding references used in the proofs.

### Mentioned references

[1] http://arxiv.org/abs/2006.13409

[2] http://dx.doi.org/10.1103/PhysRevX.10.041044

[3] https://openreview.net/forum?id=O-6Pm_d_Q-

[4] http://arxiv.org/abs/2107.14324

[5] https://arxiv.org/abs/2106.04156

[6] http://arxiv.org/abs/2006.13409



**Summary Of The Paper:**

The authors study average-case representation capacity of random ReLU neural
networks, as measured through their linear regions, as in Hanin and Rolnick
2019 -- the difference in the authors' setting is that they consider the input
space to be defined on a low-dimensional manifold, whereas Hanin and Rolnick
considered inputs on the solid cube. The authors derive analogues of each of
the results of Hanin and Rolnick in the manifold setting -- these results
include formulas for the "average volume density" of linear regions, a proxy
for the exact number of linear regions, and the average distance to the
boundary of the linear regions -- where the dimension dependence is on the
intrinsic manifold dimension rather than the ambient dimension, and additional
constants appear that depend on properties like curvatures of the manifolds.
They provide toy experiments to verify the theory (involving testing linear
region properties on a pair of 1D manifolds with different curvatures), as well
as an experiment on 'manifolds of faces' generated by the StyleGAN model -- in
both cases, the linear regions are studied throughout training rather than just
at initialization (as the theory pertains to).



**Summary Of The Review:**

The work represents a theoretical generalization of the results of Hanin and
Rolnick on linear regions in random ReLU networks, but it is an incremental
one, with proofs whose structure nearly agrees with those of the prior work.
The empirical results could be more systematic in helping to illustrate the
key novel differences between this setting and the previous setting, and why
they are important.

---

> ### Author Response · Authors · 2021-11-15
> **Response to Reviewer 1oFk**
>
> Thank you for your thorough and constructive review of the paper, and for taking the time to go through many of the details. Below we respond to all of your major concerns.
>
> On the incremental nature of our work. While we do agree, and acknowledge multiple times in our work, that we build on the work by Hanin and Rolnick we also stand by the fact that we have provided a non-trivial and insightful extension which leads to improved bounds.  As the reviewer might be very well aware of, there are various ways in which the Jacobian or the smooth co-area formula are defined. Putting them together in the right manner and incorporating the idea of planes transverse to a manifold is a unique approach for bounding of the density of linear regions. To a trained eye our contributions might seem like trivial additions to the literature but we would like to point out that these are mathematical objects that are largely missing from the theoretical analysis of DNNs. We have stuck to having propositions and statements similar to Hanin and Rolnick for ease of exposition. In some cases we do use some of the propositions proved by them directly, as is very common when building upon other theoretical works, but we have cited them for every definition or proposition we use from their work.  However, we are happy to make any specific stylistic changes you would recommend for our work.
>
>
> You are correct in pointing out that the weights of individual neurons should be rows of weights and not columns. We have changed our text accordingly; this was a minor fault on our part and does not change the results. However, can you please clarify what you mean by “the weights in the formula should be "full", rather than single neurons”? We have now pointed to the assumptions in the appendix to clarify the conditions under which our results hold right before the statement of the results. We restrict ourselves to discussing transversality in the Appendix due to space constraints. We have tried to convey the central message in the main body of the paper: that it is important to incorporate data geometry into studying the linear regions of DNNs.
>
>
> We are unable to derive similar bounds as that of Hanin and Rolnick due to the term C_{k, k} in the summation expression on the right hand side of the inequality in Theorem 3. The summation remains polynomial instead of a simpler inverse dependence that is proposed by Hanin and Rolnick. The main purpose of the result is to show that the average distance to the decision boundary depends on the average curvature of the manifold. It stands in support of Theorem 2, that the curvature of the manifold is relevant when bounding the average distance in addition to C_M. This is due the fact that distortion of the manifolds, as illustrated in Figure 10, is quantified by the curvature. We request you to please clarify the part “It seems that this result needs to be rewritten to be asymptotic for small \epsilon this should also enable a more precise conclusion to be asserted”: is the intent here that we propose an inequality which is obtained after substituting with a choice of epsilon as done by Hanin and Rolnick?
>
>
> We note that the experiments are meant to reinforce our theoretical results. By means of the circle vs tractrix example we emphasize the fact the density of linear regions is indeed different for random inilizations. We would very much appreciate suggestions as to what further interpretations you would like to see or at least in what vein. In the meantime, we have added further interpretations of the StyleGAN results in section 4.3 . Note that our intention, as stated in our work, was to study overfitting and its effects on the density of linear regions. It is intended as an observation but definitely not the main goal of our paper. Our aim for the paper is to introduce the idea of data lying on a low dimensional manifold to the study of linear regions of Deep ReLU networks.
>
>
> On “missing references to highly related work motivated similarly to the authors”. Thank you for the very relevant related work; we have added a section to the Appendix. We have also pointed to the Appendix section with related work in the main body of our paper. We have taken a two pronged approach to the related works section: listing the works that apply the manifold hypothesis in the expressive power setting and the learning dynamics setting. We would very much appreciate your comments, further additions, or clarifications.
>
> Overall, we have made changes wherever feasible as per your valuable feedback.

---

### Official Review · Reviewer_gvJZ · 2021-11-06

**Correctness:** 4
**Technical Novelty And Significance:** 4
**Empirical Novelty And Significance:** Not applicable
**Recommendation:** 8
**Confidence:** 4

**Main Review:**

The theoretical results presented here are insightful and interesting. To the extent that I could verify, they seem correct, and the empirical validation (even if a bit simplistic) demonstrates them well. The shift from Euclidean space to intrinsic manifold yields interesting differences in the results from previous work by Hanin  and Rolnick, such as eliminating exponential growth in the bounds established in the latter. It should be noted that the results here focus on capacity over unoptimized (randomly initialized) networks, and do not cover aspects in learnability. However, I believe the topics explored and exposed here are of interest in their own right, even without addressing the effects of learning or optimization of the network.

**Summary Of The Paper:**

This paper provides an analysis of the impact of data geometry on the regions of linearity in deep neural networks. To this end, it extends previous results by Hanin and Rolnick, which were established in terms of the Euclidean ambient feature space of input data, to account here for nonlinear structure when modeling the data as being sampled from a manifold submersed in this ambient space. The quantities related to the local linear structure of ReLU networks are directly related to the number of neurons, and theoretical results are validated empirically, albeit limited to simplified toy examples.

**Summary Of The Review:**

As said in the main review, this is an interesting and insightful work. Its main focus is theoretical, and clearly expands on previous work in a non-trivial way. Empirical results are somewhat simplistic, but as their main purpose is to demonstrate theoretical results this is fine in the context of this submission. Therefore, I recommend accepting it and look forward to seeing it presented in the conference.

---

### Decision · Program_Chairs · 2022-01-20

**Decision:**

Reject

**Comment:**

The paper studies the effect of manifold geometry on the complexity of the function implemented by a random ReLU network, as measured through its decomposition into linear / affine regions. In particular, it provides bounds on a surrogate for the number of such regions and the distance of a fixed point to the boundary of its region. These bounds follow from an extension of an argument of Hanin and Rolnick for Euclidean space. The bounds hold at random initialization, and are complemented with experiments in which they remain valid through training.

Initial reviews of the paper were mixed. All reviewers recognized the extension to structured / non-euclidean data as an important direction, and the results as extending the argument of Hanin and Rolnick to this setting. At the same time, there were questions about the novelty, clarity, and implications of the paper. One issue concerns the implications of the results and the amount of insight they offer into the data complexity - network complexity relationship. In particular, the paper would be stronger with a more explicit accounting for the constant C_{M,\kappa} and intuitive explanations of how manifold properties such as curvature and reach affect the number of linear regions. There were also concerns regarding the statement and proof of Theorem 3, the initial version of which only held for small \epsilon. The review also raised other smaller issues regarding the paper's clarity and implications. After considering the authors feedback and revisions, reviewers retained their mixed evaluation of the paper. This appears to be a promising direction, but a paper that could benefit from further refinement.